# A non-conserved amino acid variant regulates differential signalling between human and mouse CD28

Nicla Porciello[1], Paola Grazioli[2], Antonio F. Campese[3], Martina Kunkl[1], Silvana Caristi[1], Marta Mastrogiovanni[1], Michela Muscolini[4], Francesca Spadaro[5], Cédric Favre[6,7], Jacques A. Nunès [6] Aldo Borroto [8], Balbino Alarcon[8], Isabella Screpanti[3] & Loretta Tuosto[1]

CD28 superagonistic antibodies (CD28SAb) can preferentially activate and expand immunosuppressive regulatory T cells (Treg) in mice. However, pre-clinical trials assessing CD28SAbs for the therapy of autoimmune diseases reveal severe systemic inflammatory response syndrome in humans, thereby implying the existence of distinct signalling abilities between human and mouse CD28. Here, we show that a single amino acid variant within the C-terminal proline-rich motif of human and mouse CD28 ($P^{212}$ in human vs. $A^{210}$ in mouse) regulates CD28-induced NF-κB activation and pro-inflammatory cytokine gene expression. Moreover, this $Y^{209}APP^{212}$ sequence in humans is crucial for the association of CD28 with the Nck adaptor protein for actin cytoskeleton reorganisation events necessary for CD28 autonomous signalling. This study thus unveils different outcomes between human and mouse CD28 signalling to underscore the importance of species difference when transferring results from preclinical models to the bedside.

[1] Department of Biology and Biotechnology Charles Darwin, Laboratory Affiliated at Istituto Pasteur Italia-Fondazione Cenci Bolognetti, Sapienza University, 00185 Rome, Italy. [2] Department of Experimental Medicine, Sapienza University, 00161 Rome, Italy. [3] Department of Molecular Medicine, Laboratory Affiliated at Istituto Pasteur-Fondazione Cenci Bolognetti, Sapienza University, 00161 Rome, Italy. [4] Istituto Pasteur-Fondazione Cenci Bolognetti, 00161 Rome, Italy. [5] Confocal Microscopy Unit NMR and Confocal Microscopy Area Core Facilities, Istituto Superiore di Sanità, 00161 Rome, Italy. [6] Centre de Recherche en Cancérologie de Marseille, Immunity and Cancer Team, Institut Paoli Calmettes, Inserm U1068, CNRS, UMR7258, Aix-Marseille Université UM 105, 13284 Marseille, France. [7] Life Sciences Global Assay and Applications Development, Beckman Coulter, Immunotech SAS, 13276 Marseille, France. [8] Centro de Biología Molecular Severo Ochoa, Spanish National Research Council-Autonomous University of Madrid (CSIC-UAM), 28049 Madrid, Spain. Correspondence and requests for materials should be addressed to  L.T. (email: loretta.tuosto@uniroma1.it)

CD28 is an important co-stimulatory molecule for T lymphocytes, which delivers signals that complement T cell receptor (TCR) in both qualitative and quantitative manners, thus promoting high levels of cytokines, T cell proliferation, survival and differentiation. During T: antigen presenting cell (APC) encounter, CD28 binds to B7.1/CD80 and/or B7.2/CD86 co-stimulatory molecules, expressed on the surface of APCs (for example, macrophages, dendritic cells, and B lymphocytes), thus enhancing the close contact between T cells and APCs and mediating the actin cytoskeleton rearrangement events required for the generation of a dynamic platform at the immunological synapse, where many signalling molecules are recruited and protected from phosphatases[1]. Studies have also evidenced the ability of CD28 to function in a TCR-independent manner and to deliver unique signals regulating several biochemical events[2,3]. For instance, in the human system, CD28 stimulation by either agonistic antibodies or B7 molecules induces the recruitment of several signalling proteins[4–6] that in turn cooperate to activate a non-canonical NF-κB2-like cascade[7,8] leading to the upregulation of pro-inflammatory cytokine/chemokine genes in healthy individuals as well as in multiple sclerosis (MS) and type 1 diabetes (T1D) patients[9,10].

Until 2006, the signalling properties of CD28 between rodent (mouse and rat) and human were considered rather similar and, for several years, in vivo mouse models have been used for study the function of CD28 costimulation in health and immune diseases. Thus, when CD28 superagonistic antibodies (CD28SAb) were discovered to preferentially activate and expand immunosuppressive regulatory T (Treg) cells[11], pre-clinical experiments have been performed to evaluate the potential use of these CD28SAbs to ameliorate the onset, progression, and clinical course of human autoimmune diseases. However, when a humanised CD28SAb (TGN1412) was administered to volunteers on March 2006, the phase I clinical trial turned in a catastrophe, because this antibody induced a rapid and massive cytokine production (for example, IFN-γ, IL-1, IL-6, and TNF), thus causing a severe systemic inflammatory response syndrome[12]. Altogether, the above-reported data evidenced that the translation of experimental results from mice to humans could determine dramatic effects, thus suggesting the existence of differences in CD28 signalling capabilities between human and mouse[13,14].

CD28 signalling properties rely on the composition of its small cytoplasmic domain (41 aa), where three important motifs have been identified: one N-terminal YMNM motif and two proline-rich motifs that upon CD28 engagement induce protein recruitment through their SH2 and/or SH3 domains[2]. By comparing the sequence of the cytoplasmic tail of CD28 between human and mouse, a single amino acid variant within the C-terminal proline-rich motif was found: $P^{212}$ in human CD28 (hCD28) ($PYAPP^{212}$) vs. $A^{210}$ ($PYAPA^{210}$) in mouse.

In this study, we analyse the signalling pathways and biochemical mediators activated upon agonistic and superagonistic stimulation of primary CD4$^+$ T cells and investigate the role of the single P to A substitution within the cytoplasmic tail of human and mouse CD28. Our data provide evidence that $P^{212}$ residue within the C-terminal proline-rich motif of hCD28 is essential for delivering pro-inflammatory signals and the natural P to A substitution, in mouse CD28 can, at least in part, explain the different signalling capabilities of CD28 in human and mouse.

## Results

### CD28 pro-inflammatory signals in human but not mouse T cells.
We and others have reported that, in human primary CD4$^+$ T cells, CD28 stimulation by B7 molecules, or agonistic Abs, or CD28SAbs, in the absence of TCR ligation, induces the expression of pro-inflammatory cytokines and chemokines[7,9,10,14]. Conversely, in the mouse system, CD28 stimulation alone did not exhibit any pro-inflammatory activity[13], and CD28SAbs, instead, preferentially activated and expanded immunosuppressive Treg cells[11].

We first performed a kinetic analysis of the relevant pro-inflammatory cytokines and chemokines induced by stimulating human CD4$^+$ T cells with either agonistic (CD28.2) or super-agonistic (ANC28.1) Abs. The results reported in Supplementary Fig. 1 evidenced that almost cytokines reached a peak of expression after 6 h and returned to a basal level after 24 h from stimulation (Supplementary Fig. 1a–c). By contrast, TNF expression peaked at 1 h and returned to a basal level after 6 h from stimulation (Supplementary Fig. 1d).

The comparison of cytokine gene expression extent in CD4$^+$ T cells from human and mouse evidenced high levels of TNF, IFN-γ, IL-1β, IL-6, IL-8, and IL-2 in human CD4$^+$ T stimulated by CD28SAb compared to mouse CD4$^+$ T cells, which exhibited only a significant ($p < 0.01$, by Mann–Whitney test) upregulation of IL-2 mRNA (Fig. 1a–f), as previously reported[11,15]. The kinetic (Supplementary Fig. 2a, c, e) and dose response analysis of cytokine gene expressions (Supplementary Fig. 2b, d, f) induced by stimulating human or mouse CD4$^+$ T cells with different concentrations of crosslinked CD28SAbs revealed that, in mouse CD4$^+$ T cells, D665 Ab did not induce any significant pro-inflammatory cytokine gene expressions even at the higher dose and after 6 h from stimulation. Conversely, the lowest concentration of human ANC28.1 Ab strongly upregulated TNF, IL-6, and IFN-γ mRNAs (Supplementary Fig. 2).

Consistently with our previous findings[7,9], stimulation of human CD4$^+$ T cells with agonist CD28.2 Ab induced significant and strong increase of IL-6 ($p < 0.001$, by Mann–Whitney test) and IL-8 ($p < 0.01$, by Mann–Whitney test) mRNA levels (Fig. 1g). By contrast, stimulation of mouse CD4$^+$ T cells with agonist 37.51 Ab did not induce any significant cytokine mRNA expression (Fig. 1h). Interestingly, engagement of CD28 by its natural ligand B7.1/CD80 was also able to induce significant ($p < 0.05$, by Student's t-test) IL-8 secretion, as the agonistic CD28.2 Abs (Supplementary Fig. 3a). These data suggest that hCD28 has an intrinsic capability to trigger pro-inflammatory signals.

The higher levels of pro-inflammatory cytokines and chemokines induced by CD28 triggering in human CD4$^+$ T cells, compared to mouse CD4$^+$ T cells, could not be related to different stimulation activities of the agonist and/or super-agonist Abs. For instance, the co-engagement of CD3 and CD28 induced comparable levels of IL-2 mRNAs in both human and mouse CD4$^+$ T cells (Fig. 1i). On the contrary, a high upregulation of IFN-γ mRNA was detected only in CD3/CD28-stimulated human CD4$^+$ T cells (Fig. 1j). Moreover, super-agonistic human ANC28.1 and mouse D665 Abs efficiently induced the expansion of Treg cells (Fig. 2a). Finally, the high pro-inflammatory activity of hCD28 was not related to the preferential stimulation of effector/memory T cells, as proposed by other researchers[16,17]. Indeed, even if the mean percentage of human naïve T cells was lower than mouse (Supplementary Fig. 3b–e), no significant differences in cytokine gene expression were observed upon stimulation of human naïve or effector/memory T cells with either agonist or superagonist anti-CD28 Abs (Fig. 2b–f).

These data evidence that the pro-inflammatory CD28 behaviour is independent of the differentiation status of T cells, but is a signalling signature of the human costimulatory receptor.

**P[212] residue regulates hCD28 pro-inflammatory functions.** CD28 signalling capabilities rely on the sequence of its small cytoplasmic domain that, in both mouse and hCD28, contains important motifs that bind several signalling molecules[2]. The comparison of the sequence in the cytoplasmic tail of CD28 between human and mouse reveals a single amino acid variant within the C-terminal proline-rich motif: P[212] in hCD28 vs. A[210] in mouse (Fig. 3a). This P residue is conserved in higher primates

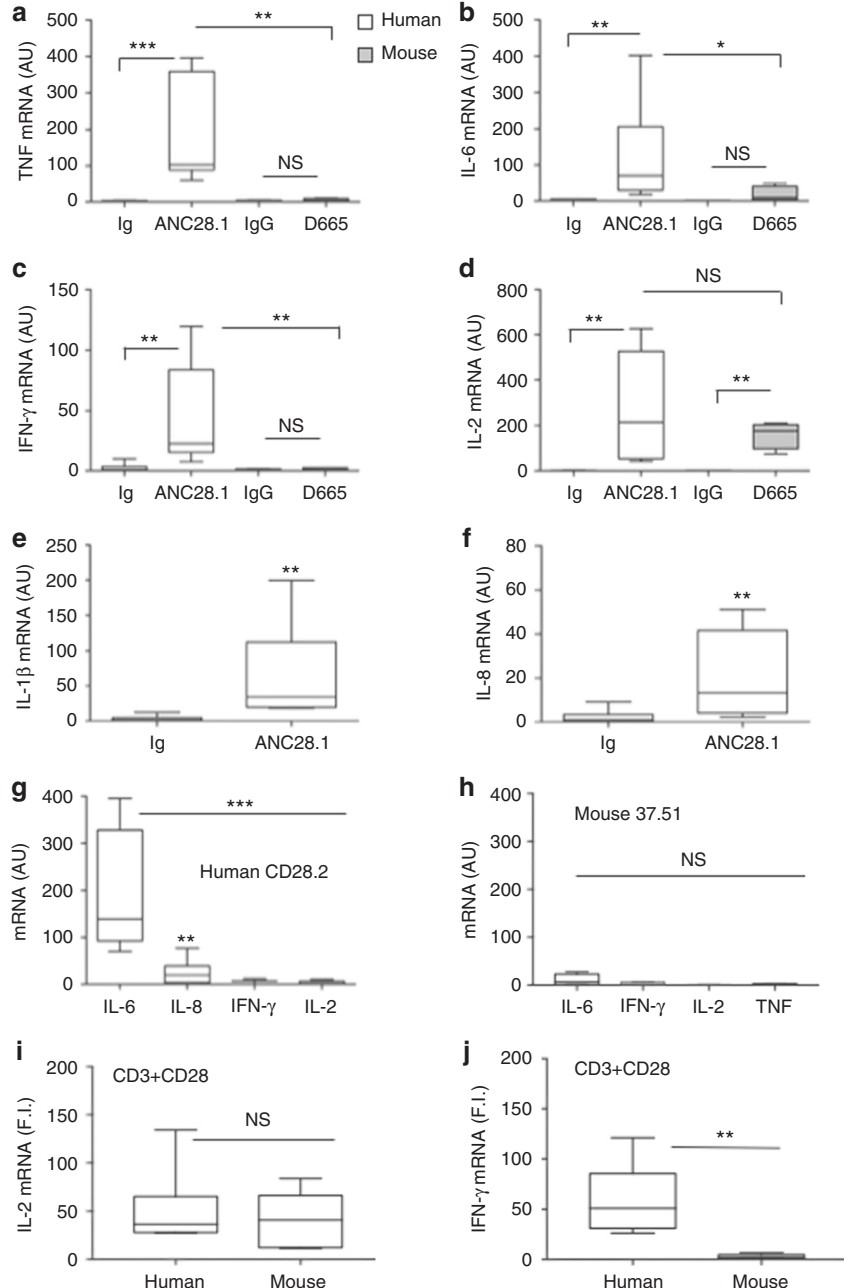

**Fig. 1** CD28 stimulation strongly upregulates pro-inflammatory cytokine and chemokine gene expressions in human but not in mouse CD4[+] T cells. mRNA levels of splenic CD4[+] T cells from $n = 13$ mice or human peripheral blood CD4[+] T cells stimulated for 1 h or 6 h with 2 µg ml[−1] isotype control mAb (Ig) or crosslinked agonistic Abs (human CD28.2 or mouse 37.51) or CD28SAbs (human ANC28.1 or mouse D665), or anti-CD3 (human UCHT1 or mouse 145-2C11) plus anti-CD28 (human CD28.2 or mouse 37.51) Abs. **a** TNF (1 h stimulation). **b** IL-6 (1 h stimulation). **c** IFN-γ (1 h stimulation). **d** IL-2 (6 h stimulation). **e** IL-1β (1 h stimulation). **f** IL-8 (6 h stimulation). Median values: TNF ($n = 7$), ANC28.1 = 102.3 vs. D665 = 2.77; IL-6 ($n = 6$), ANC28.1 = 69.7 vs. D665 = 8.5; IFN-γ ($n = 7$), ANC28.1 = 22.5 vs. D665 = 1.94; IL-2 ($n = 6$), ANC28.1 = 214 vs. D665 = 176; IL-1β ($n = 6$), ANC28.1 = 34.0; IL-8 ($n = 6$), ANC28.1 = 13.1. **g** IL-6 ($n = 8$, median = 138.9), IL-8 ($n = 9$, median = 19.5), IFN-γ ($n = 3$, median = 6.8), and IL-2 ($n = 5$, median = 1.1) mRNAs levels upon 6 h stimulation of human CD4[+] T cells with isotype control Ig or crosslinked human CD28.2. **h** IL-6 (median = 6.6), IFN-γ (median = 4.3), IL-2 (median = 0.8), and TNF (median = 2.6) mRNAs levels upon 6 h stimulation of murine CD4[+] T cells with isotype control Ig or crosslinked mouse 37.51. **i** Human IL-2 ($n = 8$, median = 36.6) and mouse IL-2 (median = 40.8), **j** human IFN-γ ($n = 7$, median = 51.1) and mouse IFN-γ (median = 2.4) mRNA levels upon 6 h stimulation of CD4[+] T cells with isotype control Ig or crosslinked anti-CD3 plus anti-CD28 Abs. All mRNAs were measured by real-time PCR and values, normalised to GAPDH (human) or RLP32 (mouse), expressed as arbitrary units (AU) or fold induction (F.I.). Lines represent median values and statistical significance was calculated by Mann–Whitney test. Asterisks indicate *$p < 0.05$, **$p < 0.01$, ***$p < 0.001$, calculated on cells stimulated with isotype control Ig

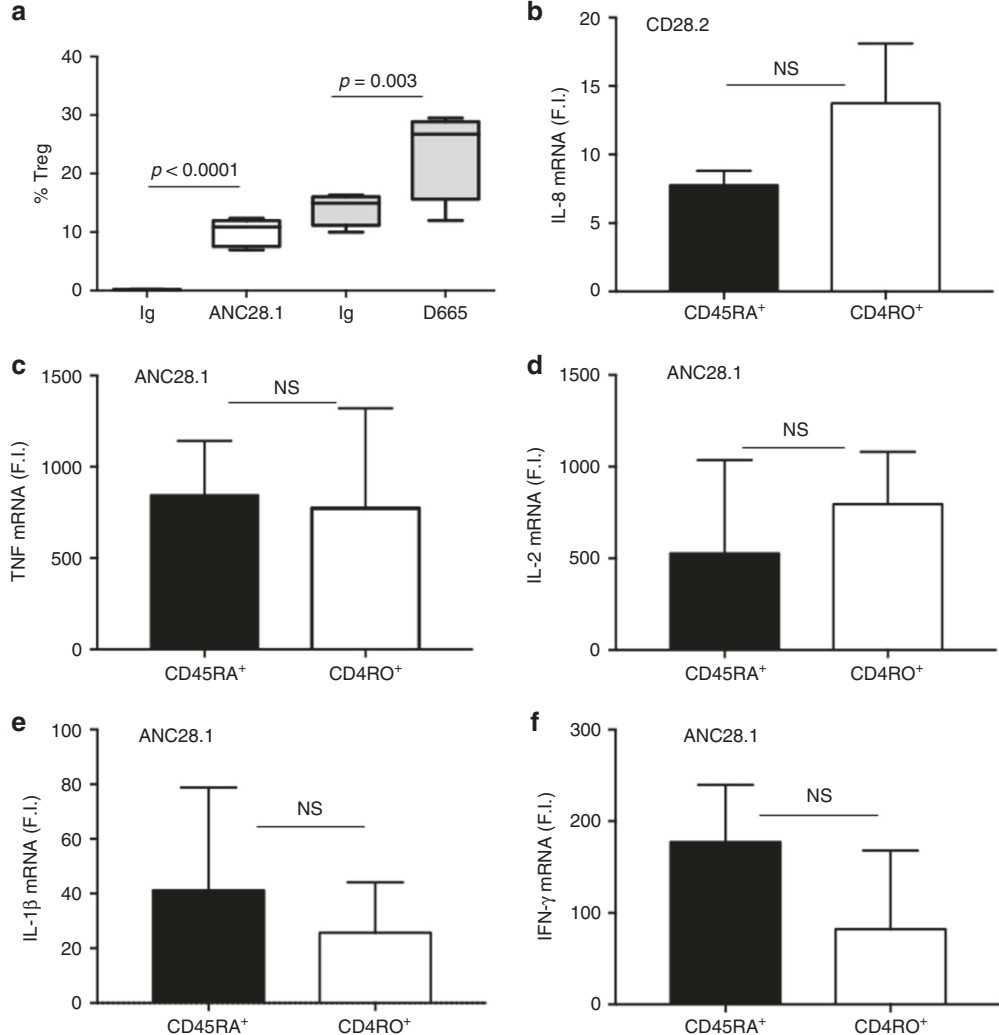

**Fig. 2** Differential effects of human and mouse CD28 stimulation in Treg cell expansion and pro-inflammatory function of naïve and effector CD4+ T cells. **a** CD4+CD25highFOXP3+ Treg expansion by 2 µg ml−1 isotype control Abs (Ig) or superagonistic anti-CD28 Abs (ANC28.1 or D665) in CD4+ T lymphocytes from human HD ($n = 5$) or mouse ($n = 4$). The percentage of CD4+CD25highFOXP3+ Treg cells were calculated. Lines represent mean values ± SD and p-values were calculated by Student's t-test. mRNA levels of IL-8 (**b**), TNF (**c**), IL-2 (**d**), IL-1β (**e**), and IFN-γ (**f**) in naïve (CD45RA+) or effector/memory (CD45RO+) CD4+ T cells from HD ($n = 3$) stimulated for 1 h (**c, e**) or 6 h (**b, d, f**) with 2 µg ml-1 of crosslinked CD28.2 (**b**) or ANC28.1 (**c–f**) Abs. Cytokine mRNA levels were measured by real-time PCR after normalisation to GAPDH (human) or RLP32 (mouse). Fold inductions (F.I.) were calculated over the basal level of cells stimulated with control Ig. Bars indicate mean F.I. ± SD. Statistical significance was calculated by Student's t-test. NS not significant

(for example, gorilla, macaque, and chimpanzee) but not in lower mammals (for example, mouse, rat, rabbit, dog, and cat).

To verify if P212 within the C-terminal proline-rich motif of hCD28 could account for the functional differences observed with respect to mouse CD28, the CH7C17 CD28-negative Jurkat T cell line (Fig. 3b) was reconstituted with hCD28 WT (hCD28WT) or hCD28 mutant containing a P212 to A substitution within the C-terminal proline-rich motif (Fig. 3a). In Jurkat cells expressing comparable levels of CD28 WT or CD28P212A mutant (Fig. 3c), cytokine mRNA levels were strongly reduced by P212A mutation following stimulation with superagonistic ANC28.1 (Fig. 3d–f) or agonistic CD28.2 Abs (Fig. 3g). Moreover, stimulation of CH7C17 Jurkat cells expressing mouse CD28 WT (mCD28WT, Supplementary Fig. 4b) with either superagonistic D665 or agonistic 37.51 Abs did not induce any significant increase of pro-inflammatory cytokines (Supplementary Fig. 4d–f). On the contrary, stimulation of murine EL-4 T cells expressing hCD28 WT (Supplementary Fig. 4c) with CD28.2 Ab strongly upregulated IL-6 (Supplementary Fig. 4g). Consistently with the pivotal

role of NF-κB[3[,7] in regulating hCD28-induced expression of pro-inflammatory cytokines/chemokines[2,9], NF-κB transcriptional activity was also strongly impaired in both mCD28WT and hCD28P212A cells following CD28 engagement by either APC (Fig. 4e, g), expressing comparable levels of human (Fig. 4b) or mouse B7.1 (Fig. 4c), or agonistic/superagonistic Abs (Fig. 4f, h) compared to hCD28WT. Interestingly, the stimulation of CH7C17 cells expressing mCD28 containing an A210 to P substitution within the C-terminal proline-rich motif (Fig. 4a, c) with agonist 37.51 (Fig. 4i, j) or superagonist D665 (Fig. 4k) Abs strongly increased NF-κB transcriptional activity (Fig. 4i), total tyrosine phosphorylation (Fig. 4j), and IL-8 mRNA levels (Fig. 4k) compared to mCD28WT, thus supporting the relevance of YAPP sequence in CD28 autonomous signalling. Finally, hCD28P212A mutant also displayed a significant ($p < 0.01$, by Student's t-test) reduction of NF-AT transcriptional activity (Supplementary Fig. 5a) and IL-2 gene expression ($p < 0.05$, by Student's t-test) following TCR/CD28 co-engagement (Supplementary Fig. 5b).

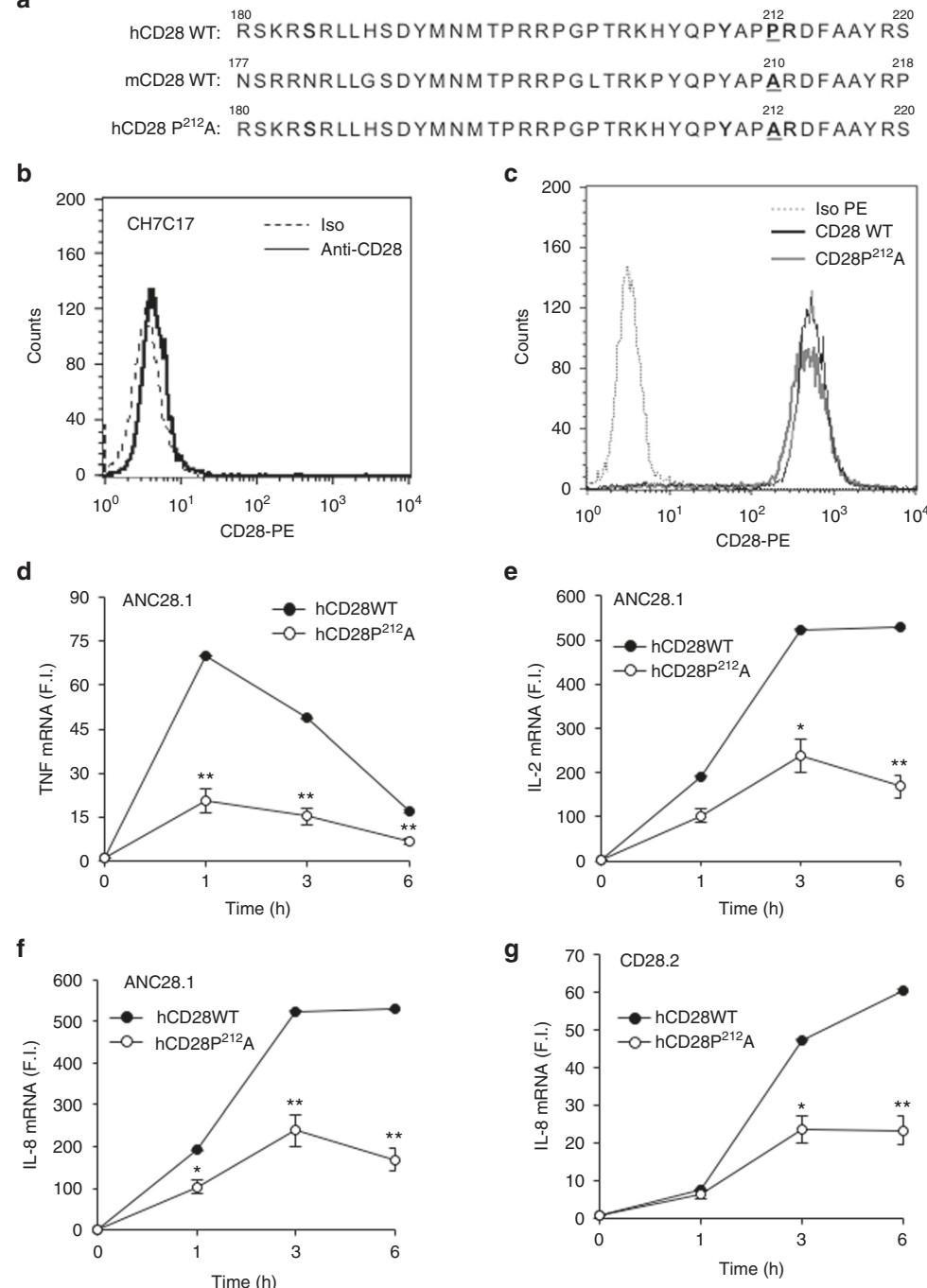

**Fig. 3** P[212] residue within the C-terminal proline-rich motif (YQPP[212]) of human CD28 is essential for CD28-mediated upregulation of pro-inflammatory cytokines. **a** Amino acid sequence of human (hCD28) and mouse CD28 WT (mCD28) and mutants. The substitution P[212]A in hCD28 is indicated in bold. **b**, **c** FACS analysis of CD28-negative CH7C17 Jurkat T cell line (**b**), or CH7C17 cells expressing hCD28WT or hCD28P[212]A mutant (**c**) stained with phycoerythrin (PE)-conjugated isotype control (Iso) or anti-CD28 (CD28-PE) Abs. **d**–**g** CH7C17 cells expressing CD28WT or hCD28P[212]A mutant were stimulated with 2 μg ml$^{-1}$ crosslinked ANC28.1 (**d**–**f**) or CD28.2 Abs (**g**). TNF (**d**), IL-2 (**e**), and IL-8 (**f**, **g**) mRNA levels were measured by real-time PCR and values, normalised to GAPDH, were expressed as fold inductions (F.I.) over the basal level of cells stimulated with control Ig. Bars show the mean ± SD of one experiment representative of three. Statistical significance was calculated by Student's $t$-test. *$p < 0.05$, **$p < 0.01$

Altogether these data suggest that the pro-inflammatory properties of hCD28 likely rely on its intrinsic capability to activate NF-κB and the absence of the P residue in the C-terminal proline-rich motif of mouse CD28 might explain the failure of mouse CD28 to induce both NF-κB activity and pro-inflammatory cytokine/chemokine gene expression.

**Human CD28 but not mouse binds to Nck**. We next proceeded to the analysis of proteins co-precipitating with CD28 in both human and mouse T lymphocytes. Based on our previous data, we specifically looked at the p85 adaptor subunit of PI3K, Vav1, and Nck[6]. An association of CD28 with Vav1 was observed in both human (Fig. 5a, upper panel) and mouse CD4$^+$ T cells

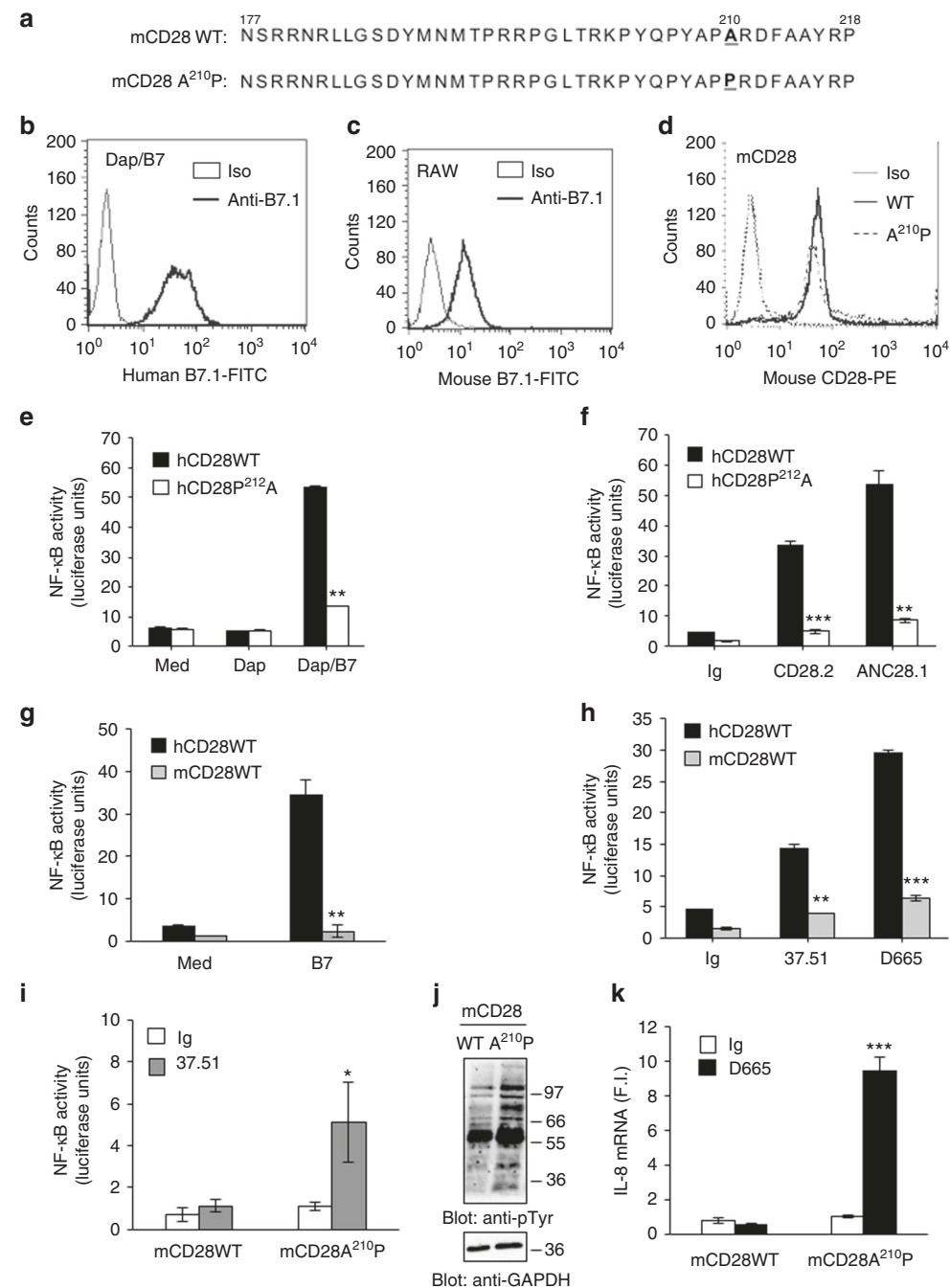

**Fig. 4** P[212]A substitution within the C-terminal proline-rich motif (YQPP[212]) of CD28 impairs NF-κB activation. **a** Amino acid sequence of mouse CD28 WT (mCD28WT) and mCD28 A[210]P mutant. The substitution A[210]P in mCD28 is indicated in bold. **b–d** FACS analysis of Dap3 cells transfected with human B7.1 (Dap/B7) (**b**) or mouse RAW 264.7 cells (Raw) (**c**) stained with FITC-conjugated isotype control (Iso) or anti-B7.1 Abs, or CH7C17 cells transfected with mouse CD28 (mCD28) WT or mCD28 A210P mutant stained with phycoerythrin (PE)-conjugated isotype control (Iso) or anti-CD28 (CD28-PE) Abs (**d**). **e–k** NF-κB luciferase activity of CH7C17 cells expressing hCD28WT (**e–h**), or hCD28P[212]A mutant (**e, f**) or mCD28WT (**g–i**) or mCD28A[210]P mutant (**i**) stimulated with B7.1 expressing cells (**e, g**) or agonistic human CD28.2 (**f**) or mouse 37.51 (**h, i**), or superagonistic human ANC28.1 (**f**) or mouse D665 (**h**). The results are expressed as the mean of luciferase units ± SD after normalisation to GFP expression. **j** Anti-phosphotyrosine (pTyr) and anti-GAPDH western blotting on total lysates of CH7C17 cells expressing mCD28WT or mCD28A[210]P mutant stimulated for 10 min with agonistic mouse 37.51 Ab. The position of molecular weight markers, expressed in kDa, is indicated on the right. **k** IL-8 mRNA levels of CH7C17 cells expressing mCD28WT or mCD28A[210]P mutant stimulated for 6 h with superagonistic mouse D665 Ab. Bars show the mean fold inductions (F.I.) ± SD, after normalisation to GAPDH. The data are representative of three independent experiments. Statistical significance was calculated by Student's *t*-test. *$p < 0.05$, **$p < 0.01$, ***$p < 0.001$, ****$p < 0.0001$

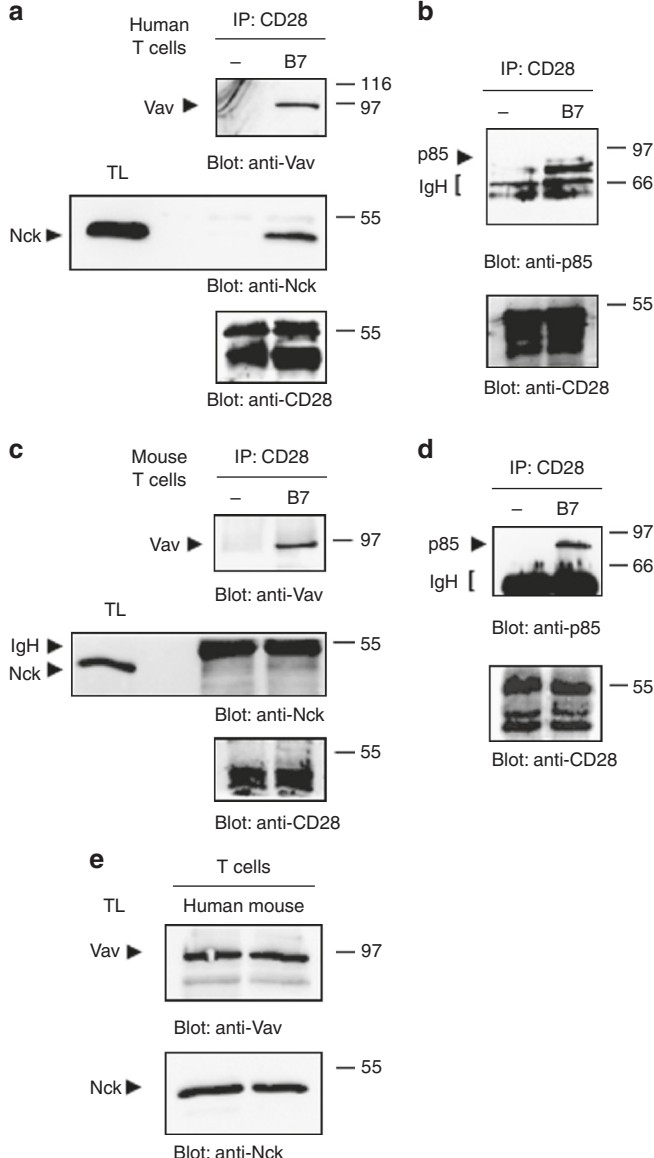

**Fig. 5** Nck binds to human but not mouse CD28. **a, b** Human primary CD4+ T cells were stimulated for 5 min with B7-negative (−) or Dap3/B7 (B7) cells and anti-CD28 IP were performed on cellular extracts. Anti-Vav, anti-Nck, anti-p85 PI3K, and anti-CD28 western blottings were performed on anti-CD28 IP or total lysates (TL). Data represent one of three independent experiments. **c, d** CD4+ T cells isolated from murine spleen (n = 10) were pooled and stimulated for 5 min with with B7-negative (−) or B7.1-expressing RAW (B7) cells and anti-CD28 IP were performed on cellular extracts. Anti-Vav, anti-Nck, anti-p85 PI3K, and anti-CD28 western blottings were performed on anti-CD28 IP or total lysates (TL). The position of immunoprecipitated proteins as well as of immunoglobulin heavy chain (IgH) is indicated. **e** Equal amounts of total lysates from human or mouse CD4+ T cells were analysed by western blotting for Vav1 (upper panel) and Nck (lower panel) content. The position of molecular weight markers, expressed in kDa, is indicated on the right

stimulated with B7-expressing cells (Fig. 5c, upper panel). On the contrary, a strong association of CD28 with Nck was observed in human (Fig. 5a, middle panel), but not in mouse CD4+ T cells (Fig. 5c, middle panel), following B7.1/CD80 engagement. Both human and mouse CD28 efficiently associated with the p85 adaptor subunit of PI3K following stimulation with B7.1/CD80-expressing

APC (Fig. 5b, d, middle panels) or with agonistic anti-CD28 Abs (Supplementary Fig. 9a). The differential association of Vav1 and Nck with human and mouse CD28 was not due to differences in the amounts of proteins, since human and mouse CD4+ T cells express comparable levels of Vav1 and Nck (Fig. 5e).

These data suggest that the binding to Nck, an adaptor molecule that plays a crucial role in the remodelling of actin cytoskeleton events regulating T lymphocyte signalling and activation[18], may contribute to the different signalling capabilities observed between human and mouse CD28.

**P[212] regulates CD28-induced actin cytoskeleton reorganisation.** Most of the CD28-mediated signalling functions rely on its intrinsic ability to regulate the remodelling of actin cytoskeleton necessary for the initiation of autonomous signalling[6,8]. Thus, we next analysed if hCD28 engagement by B7 could induce Vav1 and Nckβ recruitment and the relevance of P[212] residue. Confocal microscopy analyses revealed that hCD28 stimulation by B7.1/CD80-expressing cells (B7) induced both Vav1 (Fig. 6) and Nckβ recruitment (Fig. 7), as well as the polarisation and accumulation of F-actin at the T: APC interface (Figs. 6, 7). P[212]A substitution of hCD28 strongly impaired the recruitment of Vav1 and Nckβ as well as F-actin accumulation at the T-APC interface (Figs. 6, 7), without affecting conjugate formation (Supplementary Fig. 6b). The initial analyses of Vav1 recruitment and actin polymerisation in CH7C17 cells reconstituted with mouse CD28 (mCD28) revealed a significant reduction ($p < 0.05$, by Student's t-test) of both Vav1 recruitment and F-actin polarisation (Supplementary Fig. 6a). However, the number of conjugates between mCD28 and B7.1/CD80-expressing Raw 264.7 cells was too low to gain a statistical significance (Supplementary Fig. 6b). Thus, we generated chimeric CD28 molecules containing the extracellular and transmembrane domains of hCD28 and the cytoplasmic tail of mouse CD28 WT or mouse CD28 mutant (A[210]P), where the A within the PYAPA[210] sequence was converted to P, as the human one (YAPP[212]). In this way, the number of T: APC conjugates reached the statistical significance for performing confocal analyses (Supplementary Fig. 6b). Interestingly, the conjugation of chimera CD28 WT, containing hCD28 extracellular domain and mouse cytoplasmic domain, with cells expressing human B7.1/CD80, failed to recruit Vav1 (Fig. 6 and Supplementary Fig. 7) and Nckβ (Fig. 7 and Supplementary Fig. 7), as well as to polarise F-actin at the T: APC contact zone (Figs. 6, 7 and Supplementary Fig. 7). A210P mutation within the C-terminal proline-rich motif of mouse CD28 significantly restored Vav1 (Fig. 6 and Supplementary Fig. 7), Nckβ (Fig. 7 and Supplementary Fig. 7), and F-actin accumulation at the T: APC interface (Figs. 6, 7 and Supplementary Fig. 7).

Altogether, these data evidenced that P[212] residue within the human C-terminal proline-rich motif of CD28 is pivotal for recruiting the molecules regulating cytoskeleton reorganisation and the lack of this P residue within the C-terminal proline-rich motif of mouse CD28 may in part explain the inability of mouse CD28 to function as a TCR-independent signalling unit.

**CD28 signalling functions depend on the SH2 domain of Nck.** P[212] residue within the C-terminal proline-rich motif of hCD28 does not affect the proline consensus motif (PYAPP[212]) of CD28, but it may affect a SH2 binding consensus sequence (Y[209]APP[212]). Both Nck[18] and Vav1[19] contains a SH2 domain that may potentially bind phosphorylated YAPP sequence within hCD28 and both proteins can bind each other in a SH3-dependent manner[20]. Our ex vivo data on the loss of Nck binding to mouse CD28 (Fig. 5) strongly support a role for the SH2 domain of Nck in the recruitment of Nck, Vav1, and in F-actin

polarisation. To verify this hypothesis, we co-expressed Cherry-Vav1 together with GFP-Nckβ WT or R311Q domain or W149K mutant within the SH3.2 domain in JCH7C17 cells expressing hCD28 WT. The R311Q and W149K mutations abrogate the interaction of Nck SH2 and SH3 domains with phosphotyrosine and proline, respectively[21]. Confocal analyses revealed that both Vav1 and Nckβ were efficiently co-recruited and polarised with F-actin at the T: APC contact zone following stimulation with B7.1/CD80-positive (B7$^+$) cells (Fig. 8). On the contrary Nckβ R311Q mutant failed to recruit both Nckβ and Vav1 at the T: APC

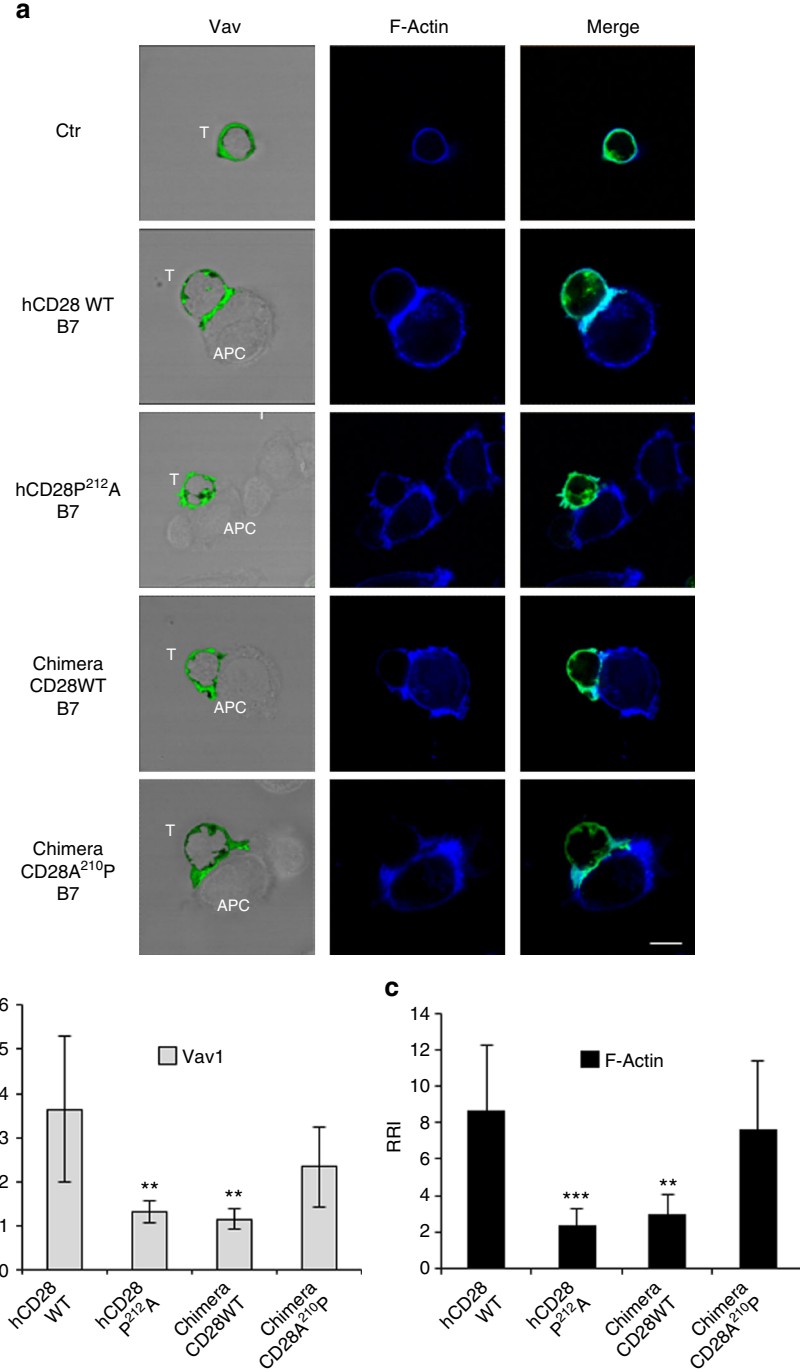

**Fig. 6** P$^{212}$ within the C-terminal proline-rich motif of human CD28 is essential for Vav1 recruitment and actin polymerisation. **a** hCD28 WT or hCD28P$^{212}$A or chimera containing the extracellular and transmembrane domain of human CD28 and the cytoplasmic tail of mouse CD28 (Chimera CD28WT) or mouse CD28 mutant (Chimera CD28A$^{210}$P) were transfected with 20 μg of GFP-Vav1 (green) and then stimulated for 15 min in the absence (Ctr) or presence of Dap/B7 cells (B7). After fixing and permeabilisation F-actin was stained with 633-conjugated phalloidin (blue) and analysed by confocal microscopy. The scale bar represents 10 μm. **b**, **c** The relative recruitment index (RRI) of Vav1 (**b**) and F-actin (**c**) was calculated as described in Methods and represents the mean ± SD of 15 conjugates analysed in each group. Mean values: Vav, hCD28WT = 3.6 ± 1.6, hCD28P$^{212}$A = 1.3 ± 0.24, Chimera CD28WT = 1.14 ± 0.2, Chimera CD28A$^{210}$P = 2.33 ± 0.9; F-actin, hCD28WT = 8.65 ± 3.6, hCD28P$^{212}$A = 2.2 ± 0.9, Chimera CD28WT = 2.9 ± 1, Chimera CD28A$^{210}$P = 7.6 ± 3.8. Asterisks **$p < 0.01$, ***$p < 0.001$ calculated by Student's $t$-test compared with hCD28WT. The results are representative of three independent experiments

interface (Fig. 8a, b) and to induce actin rearrangement (Fig. 8a, c). No differences in Vav1, Nckβ recruitment, and F-actin polarisation were observed by expressing the Nckβ W149K mutant within the SH3.2 domain (Fig. 8) or a Vav1 R[696]A mutant within the EKKAFR sequence (Fig. 9), which has been described to impair Vav1 SH2 binding ability[22]. Moreover, although exogenously expressed Vav1 (Supplementary Fig. 8a) and Nck (Supplementary Fig. 8b) could associate with hCD28 following stimulation with B7.1/CD80-expressing APC, hCD28

preferentially co-precipitated with Nck, when both Nck and Vav1 were co-expressed (Supplementary Fig. 8c).

Finally, we analysed the role of Nck recruitment and actin polymerisation in CD28 signalling functions. The overexpression of Nckβ R311K in either CH7C17 Jurkat cells expressing hCD28WT (Fig. 10a, g) or human primary CD4[+] T cells (Fig. 10b) as well as treatment with cytochalasin D (Fig. 10c–f), a known actin polymerisation inhibitor, strongly impaired CD28-induced NF-κB transcriptional activity (Fig. 10a) and pro-

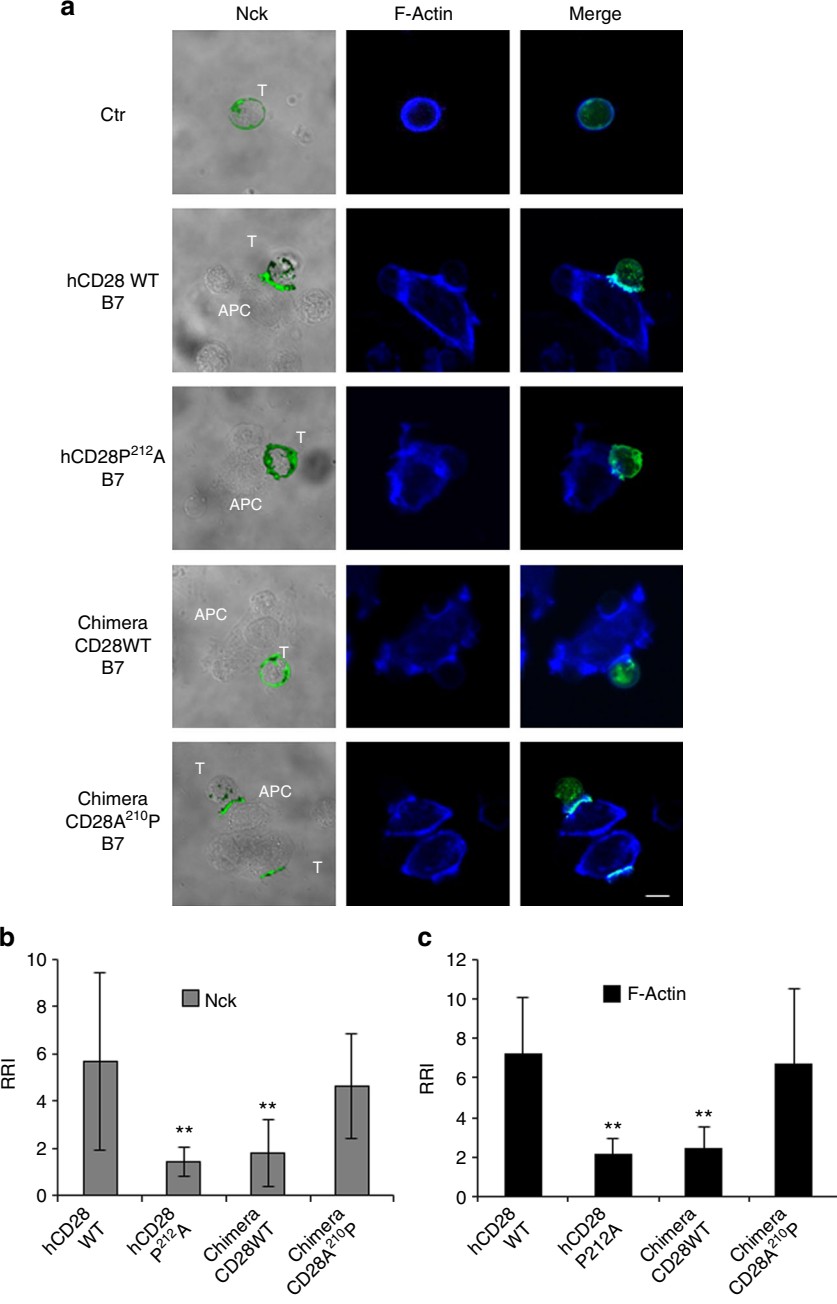

**Fig. 7** P[212] within the C-terminal proline-rich motif of human CD28 is essential for Nck recruitment and actin polymerisation. **a** hCD28 WT or hCD28P[212]A or chimera containing the extracellular and transmembrane domain of human CD28 and the cytoplasmic tail of mouse CD28 (Chimera CD28WT) mouse CD28 mutant (Chimera CD28A[210]P) were transfected with 20 μg of GFP-Nck (green) and then stimulated for 15 min in the absence (Ctr) or presence of Dap/B7 cells (B7). After fixing and permeabilisation F-actin was stained with 633-conjugated phalloidin (blue) and analysed by confocal microscopy. The scale bar represents 10 μm. **b, c** The relative recruitment index (RRI) of Nck (**b**) and F-actin (**c**) was calculated as described in Methods and represents the mean ± SD of 15 conjugates analysed in each group. Mean values: Nck, hCD28WT = 5.688 ± 3.7, hCD28[P212]A = 1.44 ± 0.6, Chimera CD28WT = 1.8 ±1.4, Chimera CD28A[210]P = 4.6 ± 2.1; F-actin, hCD28WT = 7.2 ± 2.9, hCD28[P212]A = 2.1 ± 0.8, Chimera CD28WT = 2.4 ± 1.1, Chimera CD28A[210]P = 6.7 ± 3.8. Asterisks **p < 0.01 calculated by Student's t-test compared with hCD28WT. The results are representative of three independent experiments

inflammatory cytokine expression (Fig. 10b–e), as well as CD28/TCR-induced IL-2 gene expression (Fig. 10f) and NF-AT transcriptional activity (Fig. 10g).

Altogether, these data strongly suggest that the natural P to A substitution within the C-terminal proline-rich motif of mouse CD28 causes the loss of a tyrosine binding motif (YAPP) for the SH2 of Nck, thus impairing the cooperative activity of Vav1 and Nck that is essential for optimal actin cytoskeleton rearrangement and the activation of downstream signalling pathways[23].

## Discussion

Since its discovery, it has becoming clear that CD28 may function as a TCR-independent signalling unit that delivers biochemical signals inducing several gene expression programmes in T lymphocytes. In 1997, Hünig research group described a class of superagonistic CD28-specific antibody (SAbs) able to induce, in rat, proliferation of T cells without the need for TCR engagement[24]. These non-conventional antibodies bind exclusively to the laterally exposed C″D loop of the immunoglobulin-like domain of CD28 in a parallel manner, whereas conventional, agonistic Abs recognise an epitope close to the binding site for the natural CD80/CD86 ligands, thus suggesting that different topology of bound Abs may explain their different stimulatory capabilities. The same group demonstrated that in vitro and in vivo treatment with CD28SAbs of WT mouse or mouse suffering of experimental autoimmune encephalomyelitis (EAE), an animal model for human MS, preferentially activates and expands immunosuppressive Treg cells that, upon adoptive transfer, protect recipient mice from EAE[15]. When in March 2006, after three sets of pre-clinical data, the intravenous injection of a humanised CD28SAb (TGN1412) in six healthy young men induced a strong cytokine-release syndrome[12], the entire scientific community tried to understand why the pre-clinical studies had failed to predict the cytokine storm. Despite several efforts to explain the failure of the hCD28 trial[13,14,16,17,25], the molecular players of the different outcomes between human and mouse CD28 remain still unknown. Here, we compared the molecular and biochemical events elicited following human and mouse CD28 stimulation by natural ligands, agonistic or superagonistic antibodies, thus evidencing the existence of different CD28 signalling properties between humans and mice.

One of the main differences between human and mouse CD28 relies on the ability of hCD28 to deliver TCR-independent signals regulating pro-inflammatory cytokine and chemokine gene expression[7,9,10,14]. This unique intrinsic capability of hCD28 to trigger pro-inflammatory signals is elicited by either natural B7 ligands[7] (Supplementary Fig. 3a), agonistic Abs[9], and in a more pronounced fashion by CD28SAb[14] (Fig. 1). Conversely, CD28 stimulation of mouse CD4[+] T cells with agonistic Abs does not induce any pro-inflammatory cytokine gene expression and CD28SAbs just induce proliferative IL-2 mRNA production (Fig. 1). These data are in accordance with previous results showing that, in both mice and rats, CD28SAbs induced a polyclonal IL-2-dependent T cell activation, without inflammatory adverse effects[24]. The upregulation of pro-inflammatory cytokines and chemokines by hCD28 stimulation was particularly relevant in the context of autoimmune diseases, such as MS and T1D, where we recently evidenced that CD28 stimulation by agonistic Abs strongly upregulated IL-8, IL-6, and IL-17A expression and production[9,10]. These data were in contrast to those obtained in multiple mouse models of autoimmune and inflammatory diseases, where CD28 pro-inflammatory effects were absent and CD28SAbs induced Treg cell activation and clinical improvement[11]. Moreover, the comparison of the efficacy of CD28SAb in expanding Treg cells between human and mouse

clearly demonstrate that hCD28 is equally efficient than mouse (Fig. 2a). These data are consistent with the recent observation by He et al. that CD28 stimulation by agonistic and in a stronger manner by superagonistic Abs mediates ex vivo expansion of human Treg[26]. The ability of hCD28 stimulation to expand Treg cells has been supported by several data showing that also the natural ligands B7.1/CD80 and B7.2/CD86 enhanced Treg expansion[27,28]. However, the increase of Treg cells induced by hCD28 natural ligands, agonistic, or superagonistic Abs is not able to efficiently suppress CD28 pro-inflammatory functions as evidenced by our data in both healthy, MS and T1D subject[9]. Thus, the return of CD28-superagonistic Ab TGN1412 to the clinic Phase I as TAB08[29], on the basis of the recent findings that low doses of TAB08 (0.015 μg ml$^{-1}$) increased the number of activated Treg cells without affecting pro-inflammatory cytokine production[30], must be really cautious. For instance, recent data from Weissmuller et al. evidenced that TGN1412 administration to a humanised mouse model induced strong lymphopenia, pro-inflammatory cytokine production, and death within 2–6 h[31].

One possible explanation for the strong pro-inflammatory response elicited by human CD28SAb stimulation has be related to a different balance between naïve and effector/memory T cells in human vs. mouse[16,17]. However, we did not find any significant differences in pro-inflammatory cytokine mRNA levels between naïve vs. effector/memory CD4[+] T cells upon stimulation of hCD28 with either agonistic or superagonistic Abs (Fig. 2b–f). These data are consistent with the results from Waibler et al. showing that no differences were observed in Ca$^{2+}$ release between human CD4[+]/CD45RA[+] naïve vs. CD4[+]/CD45RO[+] memory T lymphocytes stimulated with CD28SAb[14]. Thus, hCD28 pro-inflammatory functions are independent of the differentiation status of T cells, but represent a signalling signature of the human costimulatory receptor.

The signalling capability of CD28 relies on its short cytoplasmic tail that has no enzymatic activity but contains three highly conserved binding motifs (a N-terminal YMNM and two proline-rich motifs), which are crucial for the activation of downstream signalling cascade[2]. In particular, the C-terminal proline-rich motif is a master regulator of CD28 signalling functions, by binding the SH3 domain of several biochemical partners[2,32,33]. The greater role of C-terminal proline-rich motif is, however, the activation of NF-κB[3,33], and one of the main differences in the signalling functions of the C-terminal proline-rich motif between human and mouse CD28 may be related to its autonomous TCR-independent ability in activating NF-κB[2]. Although, Watanabe et al. reported that the substitution of both proline residues within the C-terminal P$^{206}$YAPA$^{210}$ motif of mouse CD28 strongly reduced TCR-mediated NF-κB transcriptional activity[34], no data on the ability of this motif to regulate CD28 autonomous signals activating NF-κB have been analysed in the mouse system. By contrast, human C-terminal P$^{208}$YAPP$^{212}$ motif mediates the recruitment of Filamin-A and associated NIK[8] to CD28, thus favouring NF-κB-dependent gene expression[6]. Herewith, we extend these data by identifying the P$^{212}$, within the C-terminal proline-rich motif of hCD28 (PYAPP$^{212}$), as a critical residue for hCD28-induced NF-κB activation (Fig. 4) and pro-inflammatory functions (Fig. 3), as well as for CD28 costimulatory signals integrating those delivered by TCR for inducing NF-AT transcriptional activation and IL-2 gene expression (Supplementary Fig. 5). Interestingly, mouse CD28 displays a natural substitution of this residue (A instead P) and lack the ability to activate NF-κB (Fig. 4h) and pro-inflammatory cytokines (Fig. 1). The reconstitution of the P residue within mouse CD28 (mCD28A210P) strongly upregulated CD28-induced NF-κB transcriptional activity (Fig. 4i), tyrosine phosphorylation (Fig. 4l), and cytokine gene expression

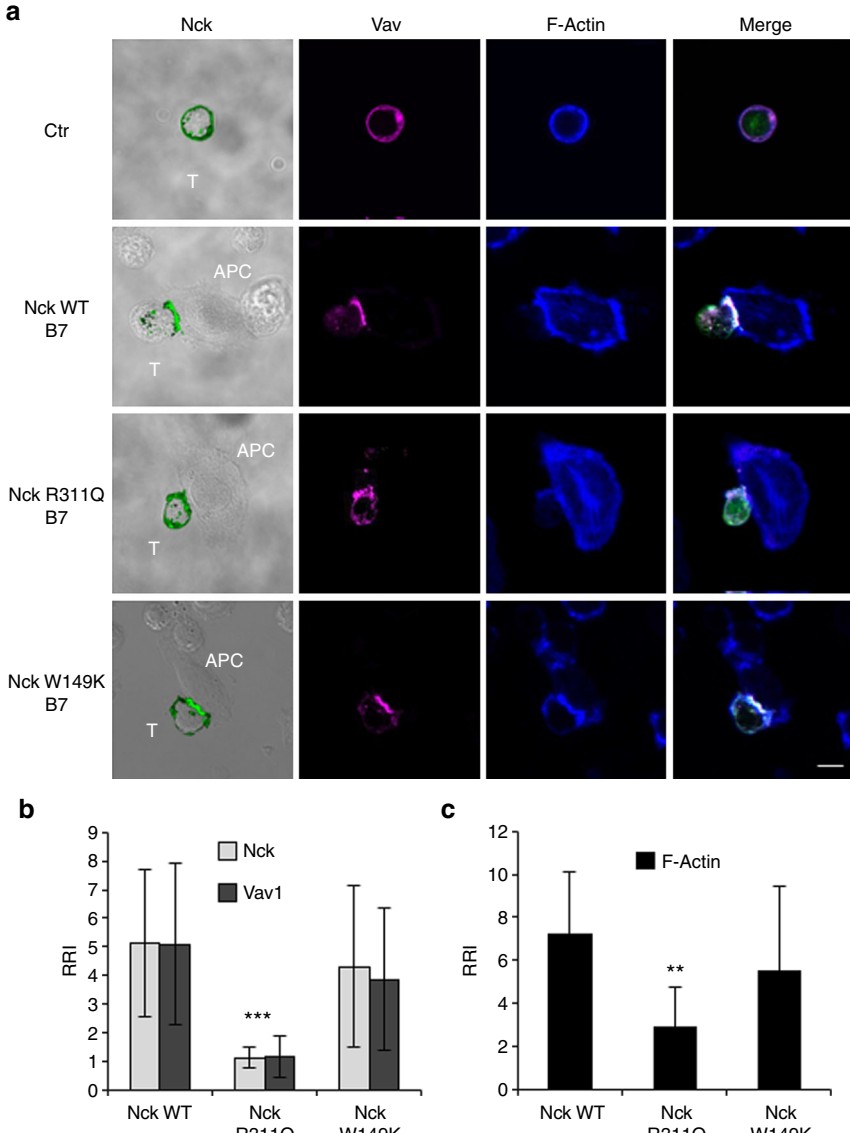

**Fig. 8** Nck SH2 domain is required for Vav1 recruitment to CD28 and actin rearrangement and polarisation. **a** hCD28WT cells were transfected with pm-Cherry Vav1 (magenta) vector together with GFP-Nck WT (green) or a GFP-Nck mutants within the SH2 domain (R311Q) or within the SH3.2 (W149K) domains and then stimulated for 15 min in the absence (Ctr) or presence of Dap3/B7 cells (B7). After fixing and permeabilisation F-actin was stained with 633-conjugated phalloidin (blue) and analysed by confocal microscopy. The scale bar represents 10 μm. The frequency of conjugates (total 15) showing significant (RRI > 2.5) Vav1 and Nck recruitment (**b**, **c**) or F-actin polarisation (**c**) were calculated and data represent the mean ± SD of three independent experiments. Mean values: Nck WT, Nck = 5.1 ± 2.6, Vav = 5 ± 2.8, F-actin = 7.2 ± 2.9; Nck R311Q, Nck = 1.1 ± 0.3, Vav = 1.1 ± 0.7, F-actin = 2.8 ± 1.8; Nck W149K, Nck = 4.3 ± 2.8, Vav = 3.9 ± 2.4, F-actin = 5.5 ± 3.9. **p < 0.01, ***p < 0.001 calculated by Student's t-test compared with Nck WT. The results are representative of three independent experiments

(Fig. 4m). Moreover, since this P residue does not affect the proline consensus motif (PYAP) of CD28, the C-terminal proline-rich motif of hCD28 likely contains other consensus sequences involved in the recruitment of pivotal signalling partners regulating its selective pro-inflammatory functions.

The results from ex vivo co-precipitation experiments in either human or mouse primary $CD4^+$ T cells stimulated with B7.1/CD80 expressing APC revealed important differences. Vav1, a Rho/Rac guanine nucleotide exchange factor, that regulates several aspect of T cell activation[35] and that we have previously found as a critical molecule coupling the C-terminal PYAPP of hCD28 to the activation of downstream signalling pathways[6,8], co-precipitated with CD28 in both human and mouse primary $CD4^+$ T cells. On the contrary, Nck, an adaptor molecule that plays a crucial role in the actin polymerisation and cytoskeleton

reorganisation processes required for cell polarisation, migration, and signal transduction[18], co-precipitates with human, but not mouse CD28 (Fig. 5). Thus, the natural P to A substitution in mouse CD28 ($YAPA^{210}$) may be responsible for the loss of a tyrosine-based consensus sequence binding for the SH2 domain of Nck. For instance, we have previously demonstrated that tyrosine residues Y207 and Y209 within the C-terminal proline-rich motif of CD28 are important for both cytoskeleton reorganisation and NF-κB activation[8]. Interestingly, recent combinatorial proteomic data from Tian et al. evidenced that Y209 together with Y191 within the YMNM motif is an abundant phosphotyrosine site and its phosphorylation depends on CD28/B7 interaction[36]. Consistently, we found that $Y^{209}F$ mutation induces a strong reduction of CD28 tyrosine phosphorylation, without affecting the tyrosine phosphorylation of Y191 as demonstrated by the

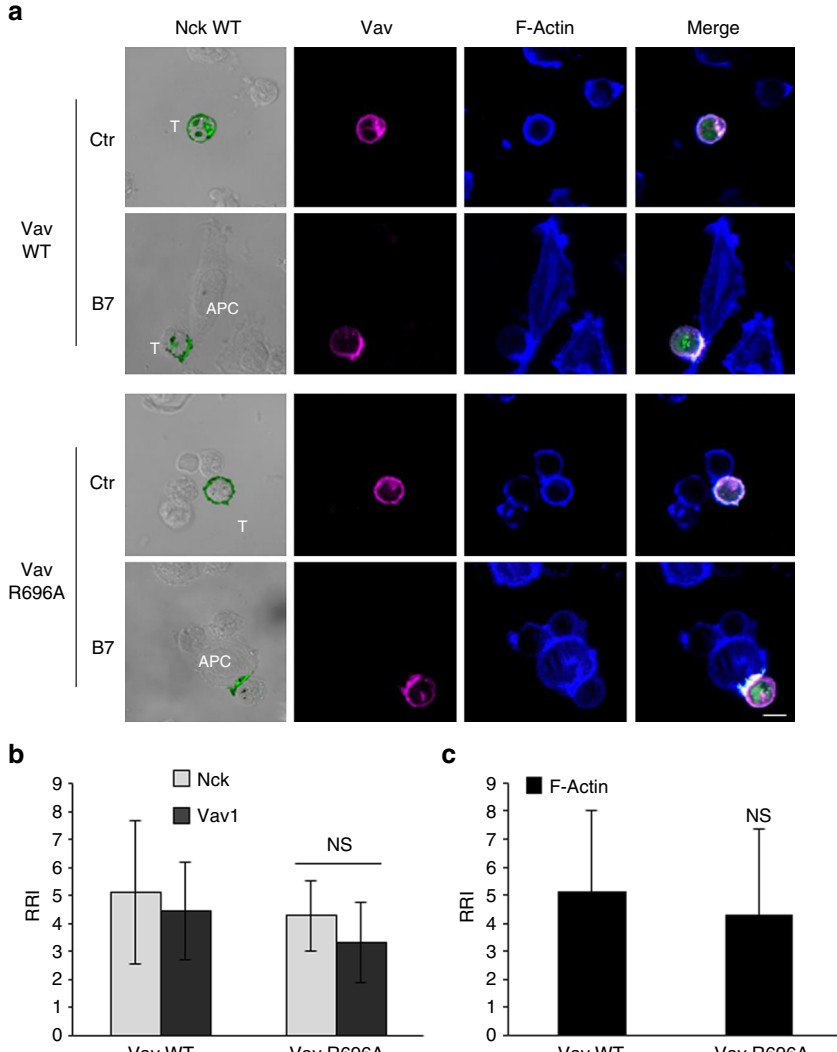

**Fig. 9** Vav SH2 domain is not required for Nck recruitment to CD28 and actin rearrangement and polarisation. **a** Human CD28WT cells were transfected with pm-Cherry Vav1 WT or Vav R696A mutant (magenta) together with GFP-Nck WT (green) and then stimulated for 15 min in the absence (Ctr) or presence of Dap3/B7 cells (B7). After fixing and permeabilisation F-actin was stained with 633-conjugated phalloidin (blue) and analysed by confocal microscopy. The scale bar represents 10 μm. The frequency of conjugates (total 15) showing significant (RRI > 2.5) Vav1 and Nck recruitments (**b**, **c**) or F-actin polarisation (**c**) were calculated and data represent the mean ± SD of three independent experiments. Mean values: Vav WT, Nck = 5.1 ± 2.5, Vav = 4.4 ± 1.7, F-actin = 7.1 ± 2.9; Vav R696A, Nck = 4.3 ± 1.2, Vav = 3.3 ± 1.4, F-actin = 6.7 ± 3.1. NS not significant

unaffected co-precipitation of p85 subunit of PI3K that depends on tyrosine phosphorylated $Y^{191}MNM$ motif[37] (Supplementary Fig. 9). However, among the 28 proteins that bind CD28 in a phosphotyrosine-dependent manner, the interactome analysis failed to detect an interaction of CD28 with Nck. One possible explanation for this discrepancy may be due to the different experimental system used in the CD28 interactome analysis. Data from Tian et al. were, indeed, obtained upon stimulation of Jurkat cells with SEE-pulsed APC, where both TCR and CD28 were co-engaged. In this experimental condition, CD28 may recruit different signalling mediators compared to CD28 stimulation alone. For instance, when TCR is engaged, Nck can be recruited to TCR by either a direct Nck association with CD3ε[38,39] or via LAT/SLP-76[23]. An interesting issue would be to compare the interaction profiles of CD28 binding proteins in the presence or absence of TCR engagement. Alternatively, a CD28-binding and Nck-binding partner may function as a linker, thus favouring CD28/Nck interaction.

Nck is composed of one SH2 and three SH3 domains involved in protein–protein interaction. More than 60 binding partners of Nck have been identified in several cell types, including Vav1. A direct association of Vav1 and Nck has been described in T lymphocytes as essential for optimal cytoskeleton remodelling[20] and Vav1/Nck complexes can be recruited to TCR by either a direct Nck association with CD3ε[38,39] or via SLP-76[23]. On the basis of the consensus sequences identified as optimal binding sites for the SH2 of Nck[18] and Vav1[19], the phosphorylated $Y^{209}APP^{212}$ within the C-terminal proline-rich motif of CD28 may likely represent an optimal binding site for Nck and Vav1. For instance, either Vav1 or Nck were able to associate with hCD28 when expressed alone (Supplementary Fig. 8a, b). However, our data on the co-precipitation of CD28 with Nck, but not Vav1, when Nck and Vav1 were co-expressed (Supplementary Fig. 8c), as well as confocal data showing that the SH2 domain of Nck, but not the SH2 domain of Vav1, was required for actin polymerisation, Nck and Vav1 recruitment (Figs. 8, 9) support a role for Nck in linking hCD28 to the cytoskeleton reorganisation events necessary for the activation of downstream signalling pathways and biological functions. For instance, an Nck R311K mutant within the SH2 domain strongly inhibited CD28-

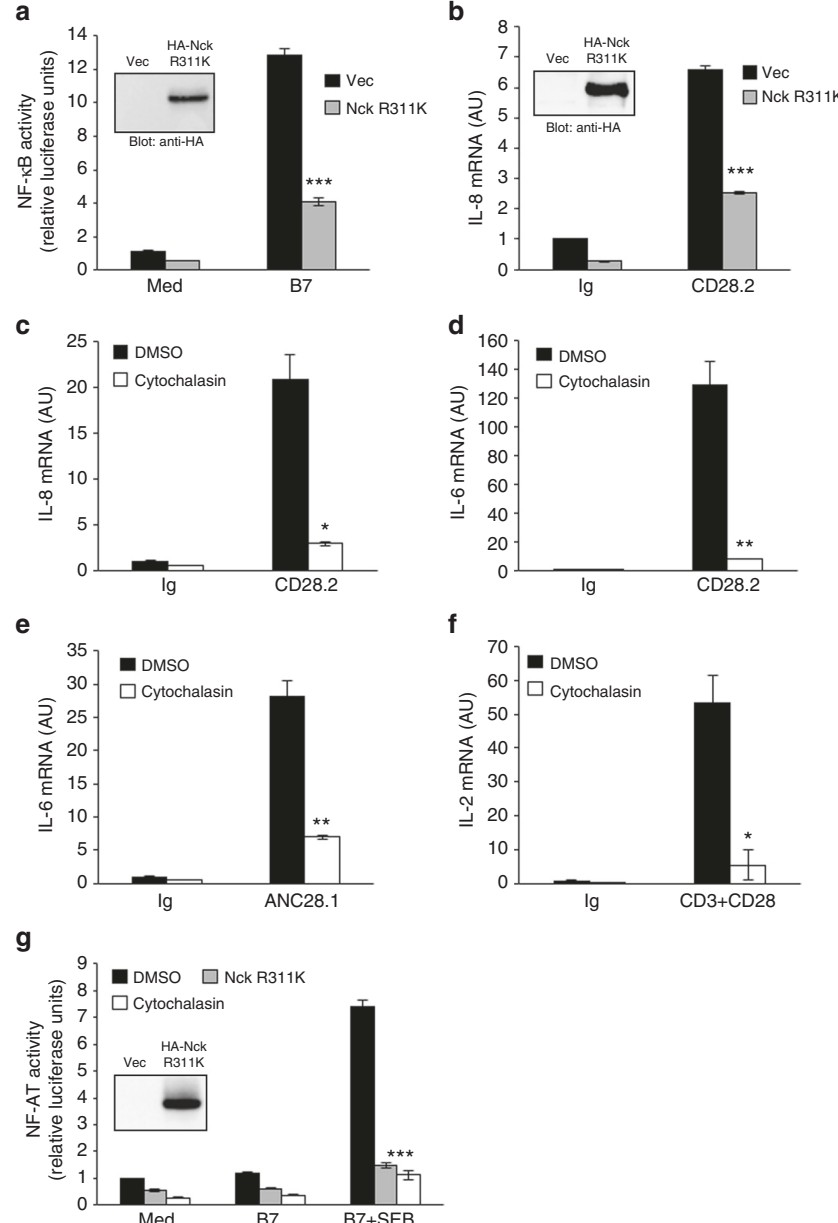

**Fig. 10** Nck SH2 domain and actin polymerisation are required for CD28-mediated signalling properties. **a** NF-κB luciferase activity of CH7C17 cells expressing hCD28WT transfected with empty vector (Vec) or HA-Nck R311K construct and stimulated in the presence (B7) or absence (Med) of Dap3/B7 cells. The results are expressed as the mean of luciferase units ± SD after normalisation to GFP expression. Asterisks (***) indicate $p < 0.001$ calculated by Student's $t$-test compared to B7-stimulated cells transfected with empty vector. Anti-HA western blotting was performed on total lysates (insert panel). The results are representative of three independent experiments. **b** Primary CD4$^+$ T cells were transfected with 2 μg of empty vector (Vec) or HA-Nck R311K construct. Real-time PCR was used to measure IL-8 mRNA levels after stimulation with isotype control (Ig) or 2 μg ml$^{-1}$ crosslinked anti-CD28.2 Abs. Data are expressed as arbitrary units (AU). Bars show the mean ± SD of four independent transfections. Anti-HA western blotting were performed on total lysates (insert panel). **c–f** Human primary CD4$^+$ T cells were stimulated for 6 h with isotype control mAb (Ig) or 2 μg ml$^{-1}$ crosslinked agonistic CD28.2 (**c**, **d**) or superagonistic ANC28.1 Abs (**e**), or 2 μg ml$^{-1}$ crosslinked anti-CD3 (UCHT1) plus anti-CD28 (CD28.2) Abs (**f**) in the presence of 10 μM cytochalasin D or DMSO, as vehicle control. IL-8 (**c**), IL-6 (**d**, **e**), and IL-2 (**f**) mRNA levels were measured by real-time PCR and values, normalised to GAPDH, expressed as arbitrary units (AU). Lines represent mean values ± SD. **g** NF-AT luciferase activity of hCD28 WT cells transfected with empty vector or HA-Nck R311K mutant and stimulated with medium (Med) or 5-3.1/B7 cells pre-pulsed or not (B7) with SEB (1 μg ml$^{-1}$) in the presence of DMSO, as vehicle control, or cytochalasin D. The results are expressed as the mean of luciferase units ± SD after normalisation to GFP values. *$p < 0.05$, **$p < 0.01$, ***$p < 0.001$ calculated by Student's $t$-test. All data are representative of three independent experiments

mediated pro-inflammatory and costimulatory signalling properties (Fig. 10). Since Nck and Vav1 are both required for the recruitment and activation of WASP/Arp2–3 complexes and actin reorganisation[23,40], the inability of mouse CD28 to recruit Nck may account for the signalling difference observed between human and mouse CD28 when engaged independently of TCR.

However, also SLP76 is required for recruitment of the Nck/Vav1-WASp complex, via the Nck SH2 domain, and actin reorganisation[41] and the cooperation of CD28 with SLP-76 and Vav1 in a TCR-independent manner has been previously described[42]. Interestingly, SLP-76 has been identified among the preferential binding proteins of CD28 PpYAPP motif[36]. Experiments are in

progress to verify this issue and to better characterise the molecular basis and the dynamic of CD28/Nck association. Moreover, we cannot exclude that other critical residues may contribute to the different CD28 signalling capabilities observed between human and mouse. Recent data from Sanchez-Lockhart et al. evidenced that, in unstimulated mouse T cells, CD28 has a weak binding affinity for B7.1/CD80[43]. Consistently, mouse CD28 exhibits a low ability to form conjugates with B7.1/CD80-expressing cells (Supplementary Fig. 6b). Since, human and mouse CD28 share only a 65% of identity in the extracellular domain, an additional contribution of the extracellular domain below human and mouse CD28 stimulation cannot be excluded.

The identification of $P^{212}$ within the C-terminal motif of hCD28 as a critical residue for CD28 pro-inflammatory and signalling functions not only opens a question for the use of rodents as model for the study of CD28-mediated functions and for the safety of new therapeutic approach, but also suggests a deeper investigation of CD28 single-nucleotide polymorphisms, some of which have been already linked to autoimmune disorders, like Bechet's disease[44] and rheumatoid arthritis[45]. Recent data from Hui et al.[46] and Kamphorst et al.[47] evidenced a dominant role of CD28 for efficient PD-1 therapy in cancer and chronic viral infections. Interestingly, a C/G single-nucleotide polymorphism (rs770610915) that causes a $P^{212}$ to $A^{212}$ missense substitution has been recently identified in hCD28. It would be interesting to analyse the presence of this polymorphism in tumour patients that do not respond to anti-PD-1/PD-Ll therapy.

## Methods

**Cells and reagents**. Human primary CD4[+] T cells were enriched from PBMCs by negative selection using a MACS microbead sorting kit (#130-096-533, Miltenyi Biotec, Milan, Italy) and cultured in RPMI 1640 supplemented with 5% human serum (Euroclone, UK), L-glutamine, penicillin, and streptomycin. The purity of the sorted population was 95–99%, as evidenced by staining with anti-CD3 plus anti-CD4 Abs (Supplementary Fig. 10). Human naïve CD45RA[+] and effector CD45RO[+] were enriched from purified CD4[+] T cell by positive and negative selection, respectively, using a MACS anti-Phycoerythrin (PE) microbeads sorting kit (#130-090-57, 1:10 dilution, Miltenyi Biotec, Milan, Italy) after labelling with a PE-conjugated anti-CD45RA primary antibody (#130-098-184, 1:10 dilution, Miltenyi Biotec, Milan, Italy). PBMCs were derived from buffy coats obtained from anonymous healthy blood (HD) donors provided by the Immunohematology and Transfusional Centre of Policlinico Umberto I (Sapienza University of Rome, Italy). Written informed consent was obtained from blood donors and both the informed consent form and procedure was approved by the Ethics Committee of Policlinico Umberto I.

Mouse primary CD4[+] T cells were enriched from spleen of C57BL/6 wild-type mice (females of 8–10 weeks) by negative selection using a MACS microbead sorting kit (#130-104-454, Miltenyi Biotec, Milan, Italy) and cultured in RPMI 1640 supplemented with 5% foetal bovine serum (FBS, Euroclone, UK), L-glutamine, penicillin, and streptomycin. The purity of the sorted population was 92–95%, as evidenced by staining with anti-CD4 Abs (Supplementary Fig. 11). Mice were bred and housed under specific pathogen-free conditions and euthanised by CO₂ inhalation. The studies involving animals have been conducted in accordance with the Italian national guidelines for use and care of experimental animals, established in D.Lgs. n.26/2014, and with European Directive 2010/63/UE. The animal protocols have been approved by the Ethical Committee of the Department of Molecular Medicine (Sapienza University of Rome, Italy). Sample sizes were chosen to assure reproducibility of the experiments in accordance with the principles of ethics regulation.

Murine EL-4 T cell line[48] was kindly provided by Polly Matzinger (National Institutes of Health, Bethesda, Maryland, USA) and was maintained as described above. CD28-negative Jurkat T cell line CH7C17[49] was kindly provided by Frederique Michel (Pasteur Institute, Paris, France) and maintained as described above with the addition of 400 µg ml⁻¹ hygromycin B and 4 µg ml⁻¹ puromycin (Sigma-Aldrich, Milan, Italy). The CH7C17 Jurkat cell line has been used because CD28-negative and easily transfectable with the CD28 mutants used in the study. CH7C17 cells, stable transfected with hCD28 WT or hCD28P²¹²A or hCD28 Y²⁰⁹FP²¹²A mutants, mCD28 WT, or chimera CD28 WT, or chimera CD28 A²¹⁰P or EL-4 cells stable transfected with hCD28 WT were generated as previously described[8]. Briefly, stable transfectants were obtained by electroporating (at 260 V, 960 µF) 10⁷ CH7C17 cells or EL-4 cells in 0.5 ml of RPMI 1640 supplemented with 20% FCS with 30 µg of constructs encoding CD28WT, or CD28 mutants. After 48 h, cells were placed in 96-well culture plates in selective RPMI 1640 medium containing 2 mg ml⁻¹ G418 (Sigma). All cell lines expressed comparable levels of CD28 (Supplementary Fig. 12). Murine L cells transfected with human B7.1/CD80 (Dap3/B7)[50], HLA-DRB1*0101 (5-3.1)[51], and 5-3.1 co-transfected with B7.1/CD80 (5-3.1/B7)[52] were kindly provided by Robert Lechler and Giovanna Lombardi (King's College London, UK). These cells have been used because several studies performed by our group evidenced that this experimental model perfectly mimics the physiological CD28/B7 encounter and stimulation in both human T cell lines and primary T lymphocytes. The mouse macrophage cell line RAW 264.7 expressing the murine B7.1 (ECACC, Salisbury, UK) was cultured in DMEM supplemented with 10% FBS (Euroclone, UK), L-glutamine, penicillin, and streptomycin.

The following antibodies were used: mouse anti-hCD28 (CD28.2, #555726, 2 µg ml⁻¹), mouse anti-human CD3 (UCHT1, #555330, 2 µg ml⁻¹), goat anti-mouse (GAM, #553998, 2 µg ml⁻¹), hamster anti-mouse CD28 (37.51, #553294, 2 µg ml⁻¹), hamster anti-mouse CD3 (145-2C11, #553057, 2 µg ml⁻¹), anti-human CD28-PE (#555729, 1:10 dilution), anti-human CD3-PE (#555333, 1:10 dilution), anti-mouse CD28-PE (#553297, 1:10 dilution), anti-mouse CD4-APC (#553051, 1:150 dilution), anti-mouse CD44-FITC (#553133, 1:100 dilution), anti-mouse CD62L-PE (#553151, 1:150 dilution) (BD Biosciences, Milan, Italy); superagonistic anti-human CD28 (ANC28.1, #177820, 2 µg ml⁻¹) (Ancell, MN, USA); superagonistic anti-mouse CD28 (D665, #LS-C58166, 2 µg ml⁻¹) (LifeSpan BioSciences, USA); rabbit anti-hamster (#A18891, 2 µg ml⁻¹ ThermoFisher Scientific, CA, USA); anti-human CD4-APC (#130091232, 1:10 dilution), anti-human CD80-PE (#130099200, 1:10 dilution); anti-mouse CD80-FITC (#104705, 1:10 dilution) (BioLegend, CA, USA); anti-human CD45RA-FITC (#21819453, 1:10 dilution), anti-human CD45RO-FITC (#21336453, 1:10 dilution) (Immunotools, Germany); goat anti-CD28 (N-20, #sc-1624, 1:400 dilution), rabbit anti-Vav (H-110, #sc-33109, 1:400 dilution), rabbit anti-HA (Y-11, #sc-805, 1:400 dilution), mouse anti-HA (F-7, #sc-7392) (Santa Cruz Biotechnology, CA); anti-PI3K pan-p85 (#06-497, 1:500 dilution), rabbit anti-Nck (#06-228, 1:500 dilution) (Merck-Millipore, Italy); anti-c-myc (9E10, #1166719001, 1:1000 dilution, Roche). Staphylococcal Enterotoxin B (SEB, BT202) was purchased by Toxin Technology Inc. (Sarasota, FL, USA) and used at 1 µg ml⁻¹.

**Plasmids cell transfection and luciferase assays**. The expression vector mouse CD28 WT (mCD28) was obtained from a PCR fragment (684 bp) cloned as a Bam HI/Not I fragment into the βCDNA4 vector (J. Nùnès, Marseille, France). The pEF-Bos construct expressing hCD28 WT (hCD28 WT) has been previously described[6]. The hCD28P²¹²A and hCD28Y²⁰⁹FP²¹²A mutants in the C-terminal proline-rich motif were derived from hCD28 WT by PCR. The hCD28P²¹²A was generated by PCR introducing the P212A substitution into hCD28WT by two-step PCR mutagenesis with the following oligonucleotides: 5'-CTATGCCCCAGCACGCGA CTTC-3' and 5'-GAAGTCGCGTGCTGGGGCATAG-3'; the hCD28Y²⁰⁹FP²¹²A was generated by PCR introducing the Y²⁰⁹F substitution into hCD28P²¹²A by two-step PCR mutagenesis with the following oligonucleotides: 5'-CATTACCAG CCCTTTGCCCCA-3' and 5'-TGGGGCAAAGGGCTGGTAATGCTT-3'.

The βCDNA4 construct expressing mouse CD28 mutant A²¹⁰P within the C-terminal proline-rich motif of mCD28 was generated by PCR introducing the A210P substitution into mCD28WT by two-step PCR mutagenesis with the following oligonucleotides: 5'-CTACGCCCCTCCCAGAGACTTTG-3' and 5'-CAAAGTCTCTGGGAGGGGCGTAG-3'. The chimeric CD28 molecules containing the extracellular and transmembrane domain of hCD28 and the cytoplasmic tail of mouse CD28 WT (chimera CD28 WT) or mouse CD28 mutant A²¹⁰P (chimera CD28 A²¹⁰P) were generated as described by PCR and cloned in BamH1 and Not I restriction sites of βCDNA4 vector. The following oligonucleotides were used: 5'-CCTGGATCCAGGACAAAGATGCTCAGGCTG-3',

5'-ACAGTGGCCTTTATTATTGTTATCTGGACAAATAGTAGAAGGAACA GA-3', 5'-TCTGTTCCTTCTACTATTTGTCCAGATAACAATAATAAA GGCCACTGT-3', and 5'-GAGGTGCCGTAAAGCACTAAATCGGAAC-3'. The entire sequence of the mutants was verified by DNA sequencing. Stable transfectants were obtained by electroporation (at 260 V, 960 µF) 10⁷ CH7C17 cells in 0.5 ml of RPMI 1640 supplemented with 20% FCS with 30 µg of pEFBos-neo encoding hCD28 WT, or hCD28 mutants or βCDNA4 encoding mCD28 WT. After 48 h, cells were placed in 96-well culture plates in selective RPMI 1640 medium containing 2 mg ml⁻¹ G418 (Sigma-Aldrich). Transfectants were subjected to cell sorting by FACScan (BD Biosciences, San Jose, CA) to obtain stable cell lines expressing similar levels of CD28.

pEF-Bos expressing C-terminal myc-tagged Vav1 was previously described[6]. The substitution of Vav1 (R696A) mutant was derived from C-terminal myc-tagged Vav1 by PCR. The oligonucleotides used were: 5'-ACTTTCTTGG TGGCGCAGAGGGTGAAG-3' and 5'-CTTCACCCTCTGCGCCACCAAG AAAGT-3'. All oligonucleotides were purchased by Eurofins Genomics (Edersberg, Germany).

Cherry and GFP constructs were generated by cloning Vav1 and Vav1 R696A mutant in Eco-RI and Bam HI restriction sites of pm-Cherry-C1 and pEGFP-N1 expression vectors (Clontech, UK), respectively. The sequence of Vav1 constructs were verified by DNA sequencing. The NF-κB luciferase gene under the control of six thymidine kinase NF-κB sites[53] was kindly provided by J.F. Peyron (Centre Méditerranéen de Médecine Moléculaire, Nice, France). The NF-AT luciferase reporter construct containing the luciferase gene under the control of the human IL-2 promoter NF-AT binding site[54] was kindly provided by C. Baldari (University of Siena, Siena, Italy).

GFP-Nck WT, GFP-Nck R311Q, and GFP-Nck W149K mutant constructs were provided by A. Borroto (Madrid, Spain). HA-Nck WT and HA-Nck R311K mutant constructs were provided by Wei Li (University of Southern California, Los Angeles, CA, USA).

Primary CD4[+] T cells, suspended in 100 μl of Nucleofector solution (VPA-1002, Amaxa Biosystems), were electroporated with 1 μg of the indicated expression vector using the V-024 programme of the Nucleofector.

For luciferase assays, $10^7$ cells were electroporated (at 260 V, 960 μF) in 0.5 ml RPMI 1640 supplemented with 20% FCS with 2 μg NF-κB luciferase or 10 μg NF-AT luciferase together with 5 μg pEGFP and each indicated expression vector, keeping the total amount of DNA constant (40 μg) with empty vector. 24 h after transfection, cells were stimulated with Dap3 or Dap3/B7 or 5-3.1 or 5-3.1/B7 cells pre-pulsed with SEB (1 μg ml$^{-1}$) at 37 °C for 6 h. Luciferase activity was measured according to the manufacturer's instruction (#E1500, Promega). Luciferase activity determined in triplicates was expressed as arbitrary luciferase units after normalisation to GFP values.

**Immunprecipitations and immunoblotting.** Primary CD4[+] T cells and Jurkat cells were stimulated as indicated at 37 °C. At the end of incubation, cells were harvested and lysed for 30 min on ice in 1% Nonidet P-40 lysis buffer (150 mM NaCl, 20 mM Tris-HCl (pH 7.5), 1 mM EGTA, 1 mM MgCl$_2$, 50 mM NaF, 10 mM Na$_4$P$_2$O$_7$) in the presence of inhibitors of proteases and phosphatases (10 μg ml$^{-1}$ leupeptin, 10 μg ml$^{-1}$ aprotinin, 1 mM NaVO$_4$, 1 mM pefablock-SC). Extracts were precleared for 1 h with Protein-A (Amersham) or Protein G Sepharose beads (Sigma-Aldrich) and then immunoprecipitated for 2 h with the human anti-CD28 Abs (CD28.2, 2 μg per immunoprecipitation) pre-adsorbed on Protein-G Sepharose beads (20 μl per immunoprecipitation) or rabbit anti-hamster Abs (2 μg per immunoprecipitation) pre-adsorbed on Protein-A Sepharose beads (20 μl per immunoprecipitation) further incubated with mouse anti-CD28 (37.51, 2 μg per immunoprecipitation). Proteins were resolved by SDS-PAGE and blotted onto nitrocellulose membranes. Blots were incubated with the indicated primary antibodies, extensively washed and after incubation with horseradish peroxidase (HRP)-labelled goat anti-rabbit (#NA934V, 1:5000 dilution) or HRP-labelled goat anti-mouse (#NA931V, 1:5000 dilution) (Amersham), or (HRP)-labelled donkey anti-goat Abs (#sc-2033, 1:5000 dilution) (Santa Cruz Biotechnology) developed with the enhanced chemiluminescence's detection system (GE Healthcare). Uncropped version of all Western blots can be found in Supplementary Figs. 14 and 15.

**Cytokine production.** Secretion of IL-8 was measured from the supernatants of CD4[+] T cells cultured for 24 h in flat bottom 48 culture wells (3 × 10$^5$ cells per well) either unstimulated or stimulated with crosslinked anti-CD28.2 Ab (2 μg ml$^{-1}$) or adherent Dap3/B7 cells fixed with paraformaldehyde to avoid detachment and secretion. A cytometric bead-based immunoassay (#558277, CBA FLEX SET, BD Biosciences) was used to measure IL-8. Data were analysed with the FCAP Array v3.0.1 software (Soft Flow Hungary Ltd.).

**Real-time PCR.** Total RNA was extracted using Trizol (Thermo Fisher Scientific, CA, USA) from 2 × 10$^6$ cells and RNeasy MicroKit (#74004, Qiagen) from 5 × 10$^5$ cells according to the manufacturer's instructions and was reverse-transcribed into cDNA by using Moloney murine leukaemia virus reverse transcriptase (Invitrogen). TaqMan Universal PCR Master Mix, human: IL-8, IL-2, IL-6, IFN-γ, TNF, IL-1β and endogenous GAPDH, mouse: IL-2, IFN-γ, TNF and endogenous RLP32 primer/probe sets were purchased from Applied Biosystems. The relative quantification was performed using the comparative $C_T$ method. The median value of human or mouse CD4[+] T cell stimulated with control isotype matched Abs was used as $C_T$ calibrator in all comparative analyses.

**Confocal microscopy.** 15 × 10$^3$ murine Dap3/B7 or RAW 264.7 cells were adhered on cover glasses (12 mm) overnight at 37 °C. CH7C17 cells expressing hCD28 WT, or mCD28 WT, or CD28P$^{212}$A mutant, or chimera CD28 WT, or chimera CD28 A$^{210}$P mutant were transfected for 24 h with 20 μg of GFP or cherry constructs, seeded on cover glasses for 15 min at 37 °C, fixed by 2% paraformaldehyde and permeabilised by 0.1% saponin in PBS containing 1% BSA. Filamentous actin (F-Actin) was stained by using phalloidin-633 (#A22284, 1:75 dilution, Molecular Probes). Confocal observations were performed using a Leica DMIRE (Leica Microsystems, Heidelberg, Germany) and a Zeiss LSM 780 (Zeiss, Berlin, Germany). Images were analysed with the Adobe® Photoshop® programme. The relative recruitment index (RRI) was calculated as previously described by the formula: RRI = [mean fluorescence intensity (MFI) at synapse−background]/[MFI at all the cell membrane not in contact with APC−background]. At least 15 cells or conjugates were examined quantitatively for each experiment. Statistical significance was calculated using a Student's t-test. Signals from different fluorescent probes were taken in parallel. Several cells were analysed for each labelling condition, and representative results are presented.

**Measurement of conjugate formation.** Conjugate formation was measured as previously described[6]. Briefly, CH7C17 Jurkat cells expressing hCD28 WT, or mCD28 WT or CD28 mutants were transfected with pEGFP construct and transfectants (3.5 × 10$^6$) were incubated for 5 min at 37 °C with Dap3/B7 (1.2 × 10$^6$) or RAW-264.7 cells in a final volume of 70 μl of RPMI, then diluted in 500 μl RPMI and analysed by FACS. Conjugates were identified on a total of 10$^5$ GFP-positive events by gating for SSC and FSC and expressed as mean percentage ± SD of triplicate samples. An example of the gating strategy used is reported in Supplementary Fig. 13.

**Statistical analysis.** The sample size was chosen based on previous studies to ensure adequate power. Parametrical statistical analysis (mean and SD) was performed to evaluate differences between continuous variables through Prism 5.0 (GraphPad Software, San Diego, CA) using standard unpaired t-test. For multiple group comparisons, significant differences were calculated using the nonparametric Mann–Whitney U test, and linear regression analyses were performed using the Pearson chi-squared test. For all tests, p-values < 0.05 were considered significant.

**Study design.** Sample sizes were determined based on similar previous studies. No samples or animals were excluded from analysis. No blinding or randomisation was used during the experiments.

**Data availability.** The data that support the findings of this study are available from the corresponding author upon reasonable request.

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

## Acknowledgements

The work was supported by: "Progetto Ateneo" (Sapienza University of Rome, Italy) and Multiple Sclerosis Italian Foundation (FISM 2016/R/29).

## Author contributions

N.P. performed the experiments, analysed the data, and contributed in writing; P.G., A.F.C., M.K., M.M., F.S., M.M. performed the experiments; C.F., A.B., J.A.N., B.A. provided critical reagents and helped edit the manuscript; S.C. generated all constructs described in the paper; I.S. contributed to design the experiments, discussions and agreement with the conclusions presented; L.T. designed the experiments and wrote the manuscript.

## Additional information

**Competing interests:** The authors declare no competing interests.

