## [Peer Review File · Nature Communications]

Reviewers' comments:

Reviewer #1 (TCR signaling)(Remarks to the Author):

In this manuscript, the authors have shown that proline 212 (P212) of human CD28 protein is crucial for CD28-induced inflammatory cytokine production in human T cells. Mutating proline 212 to alanine, which represents a natural variant in mouse, largely attenuates CD28-induced cytokine production. Mechanistically, the authors suggest that P212 is critical for recruiting Nck to CD28 and promote actin polymerization upon B-7/CD28 ligation in human T cells. This unique ability of recruiting Nck to human CD28 may contribute to the differential signaling between human and mouse CD28.

Major concerns:

1. In the first two figures, the authors demonstrated differential expression of CD28-induced cytokines between human and mouse T cells from different tissues (blood vs spleen). Moreover, in the human and mouse cell experiments, the stimulating antibodies are different mAbs from different species. The differential effects observed by the authors may be due to different antibody affinities and binding epitopes. A more convincing experiment would be to compare the signaling of human CD28 with chimeric CD28, where the extracellular domain of human CD28 is fused to the intracellular domain of mouse CD28.

2. The authors have shown that P212A mutation in human CD28, which represents the natural variant in mouse CD28, attenuates CD28-induced cytokine production in Jurkat cells. But there are other sites different between human and mouse CD28 intracellular domains which may also contribute to differential CD28-induced signaling that the authors completely ignore. It would be important to know if the A210P mutation in mouse CD28 is a gain-of-function.

3. The authors have shown that P212 in human CD28 is important to recruiting Nck and inducing actin polymerization. But the authors didn't provide evidence showing either that Nck or actin regulates P212 site mediated signaling. It is important to know if P212 dependent CD28-induced cytokine production diminishes in Nck deficient cells or in the presence of actin polymerization inhibitors.

4. The authors have shown that human but not mouse CD28 recruits Nck. However, it has not been demonstrated that a defect in Nck recruitment results in defective cytokine/chemokine upregulation. Experiments using Nck mutants should be performed to link the defect in Nck recruitment to the defect in NfKB activation as well as chemokines/cytokines.

5. Furthermore, in Figures 1 and 2, mRNA levels of cytokine and chemokines are assessed only at 1 and 6 hours post stimulation. Measurement of chemokine/cytokine levels across longer time points of stimulation (i.e. 12 and 24 hours) should be performed to determine whether there are kinetic differences in their up-regulation. In fact, it is quite surprising to see IL-2 mRNA upregulation at 1 hour.

Reviewer #2 (T cell activation)(Remarks to the Author):

This is a very interesting manuscript that suggests that the single amino acid difference between mouse and human CD28 is responsible for the ability of human CD28 to induce pro inflammatory

cytokine production. The mechanism appears to be mediated by the distinct binding of Nck and Vav to the conserved proline motif. Surprisingly, Nck binds via its SH2 domain, while Vav is assumed to be recruited by its SH3 domains.

In general, I found the hypothesis to be intriguing but too preliminary in its present form to be conclusive.

1. It would be important to show that human CD28 expressed in a mouse T cell induces pro-inflammatory cytokines and vice versa. This is important to establish the basic phenomenon as the alternative hypothesis is that this can be explained by differences in the antibody.
2. Reconstitution of the CD28 negative Jurkat does not show the level of CD28 expression in the untransfected cell line, only the antibody control is shown.
3. If this is a specific effect on pro inflammatory cytokine, why is there also a defect in NFAT and IL2 production?
4. Again, the data showing distinct complexes formed with human vs mouse CD28 needs more work. Vav binds to both mouse and human but GRB2 binds only to mouse and not human. Lastly, Nck binds to human but not mouse. Does the addition of proline to mCD28 recapitulate the phenotype, meaning loss of GRB2 binding and addition of NCK binding in mouse cells?
5. The images in figures 6 is not completely convincing. The image of the chimera CD28A210P looks like minimal recruitment. Why doesn't it reconstitute similar to human if this is the only difference in the sequence? The NCK recruitment images look much more convincing.
6. The idea that CD28 binds to the NCK SH2 domain would require that the tyrosine in the proline motif be phosphorylated. Is there any evidence for this? Also, the data suggests that Vav and NCK are competing for the same binding site. Could differences in the amount of both proteins in mouse and human cells be a potential explanation for differential binding? The assumption is that this mediated by the VAV SH3 domain but this isn't shown. Another important control would be to show that the SH3 domains of NCK are "not" mediating the recruitment.

Point by point reply to reviewers' comments

We are grateful to the reviewers for their suggestions and advices to improve several aspects of the manuscript. We have carefully addressed their critical comments and we hope that our corrections satisfy their requests.

Reviewer #1:

Point 1.

Reviewer: *In the first two figures, the authors demonstrated differential expression of CD28-induced cytokines between human and mouse T cells from different tissues (blood vs spleen). Moreover, in the human and mouse cell experiments, the stimulating antibodies are different mAbs from different species. The differential effects observed by the authors may be due to different antibody affinities and binding epitopes. A more convincing experiment would be to compare the signaling of human CD28 with chimeric CD28, where the extracellular domain of human CD28 is fused to the intracellular domain of mouse CD28.*

Response: We agree with the reviewer that mAbs from different species may exhibit differential antibody affinities and binding epitopes. However, the anti-CD28 mAbs used in human and mouse cell experiments recognize similar epitopes. Indeed, agonistic anti-human CD28.2 and anti-mouse 37.51 recognize an epitope close to the binding site for the natural CD80/CD86 ligands, whereas superagonistic human ANC28.1 and mouse D665 SAb bind to the laterally exposed C'D loop. The differential effects observed on pro-inflammatory cytokine expression between the two species have been also observed by other groups (see references N. 7, 8, 9, 10, 11, 13, 14) and cannot be related to different antibody affinities, since we observed that agonistic human CD28.2 and mouse 37.51 were able to co-stimulate TCR-induced IL-2 expression at a similar extent (see new Figure 1i). Moreover, human ANC28.1 and mouse D665 SAb equally expand Treg cells (Fig. 2a). To further evaluate the stimulating properties, we also looked at the ability of human and mouse CD28 to recruit the p85 regulatory subunit of class 1A PI3K following stimulation with either CD28.2 or 37.51 agonistic Abs and we found a similar extent of recruitment (see new supplementary Figure 9a). These data have been better discussed in the results section. Finally, to confirm that the *ex-vivo* data in primary human and mouse T cells were related to different qualitative signaling abilities of CD28, we generated a mouse CD28 mutant containing an A210P substitution to restore the human YAPP sequence within the C-terminal proline rich motif of mouse CD28. The stimulation of CH7C17 Jurkat cells expressing mCD28A210P with agonistic 37.51 or superagonistic D665 Abs (new Fig. 4), strongly increased NF- κ B transcriptional activity (Fig. 4i), tyrosine phosphorylation (Fig. 4l) and cytokine gene expression (Fig 4m).

Point 2.

Reviewer. *The authors have shown that P212A mutation in human CD28, which represents the natural variant in mouse CD28, attenuates CD28-induced cytokine production in Jurkat cells. But there are other sites different between human and mouse CD28 intracellular domains which may also contribute to differential CD28-induced signaling that the authors completely ignore. It would be important to know if the A210P mutation in mouse CD28 is a gain-of-function.*

Response: As specified before (point 1), we restored the human YAPP sequence within the C-terminal proline rich motif of mouse CD28, by generating a mCD28 mutant containing the A210P substitution. The stimulation of CH7C17 Jurkat cells expressing mCD28A210P with agonistic 37.51 or superagonistic D665 Abs (new Fig. 4), strongly increased NF- κ B transcriptional activity (Fig. 4i), tyrosine phosphorylation (Fig. 4l) and cytokine gene expression (Fig 4m).

Point 3.

Reviewer: *The authors have shown that P212 in human CD28 is important to recruiting Nck and inducing actin polymerization. But the authors didn't provide evidence showing either that Nck or actin regulates P212 site mediated signaling. It is important to know if P212 dependent CD28-induced cytokine production diminishes in Nck deficient cells or in the presence of actin polymerization inhibitors.*

Response: We thank the reviewer for helpful suggestions and according to them, we stimulated either CD28 WT Jurkat cells and human primary CD4⁺ T cells with B7, or agonistic CD28.2 or superagonistic ANC28.1 Abs in the presence or absence of cytochalasin D, a known inhibitor of actin polymerization. The results shown in the new Figure 10 of the revised manuscript, clearly demonstrate a pivotal role of actin polymerization in CD28-induced pro-inflammatory cytokine gene expression (Fig. 10c-e) as well as in CD28-mediated signaling regulating TCR/CD3-induced IL-2 gene expression (Fig. 10f) and NF-AT transcriptional activity (Fig.10g). Altogether these data demonstrate a pivotal role of actin polymerization in CD28-mediated signaling.

Point 4.

Reviewer: *The authors have shown that human but not mouse CD28 recruits Nck. However, it has not been demonstrated that a defect in Nck recruitment results in defective cytokine/chemokine upregulation. Experiments using Nck mutants should be performed to link the defect in Nck recruitment to the defect in NfKB activation as well as chemokines/cytokines.*

Response: According to the reviewer's suggestions, we overexpressed Nck R311Q mutant in either CD28 WT Jurkat cells or primary CD4⁺ T cells and we analysed CD28-induced NF-κB activity (new Fig. 10a) and IL-8 gene expression (new Fig. 10b). The strong impairment of both NF-κB (Fig. 10a) and IL-8 gene expression (Fig. 10b) induced by Nck R311Q mutant demonstrate a pivotal role of Nck recruitment in CD28-mediated signaling.

Point 5.

Reviewer: *Furthermore, in Figures 1 and 2, mRNA levels of cytokine and chemokines are assessed only at 1 and 6 hours post stimulation. Measurement of chemokine/cytokine levels across longer time points of stimulation (i.e. 12 and 24 hours) should be performed to determine whether there are kinetic differences in their up-regulation. In fact, it is quite surprising to see IL-2 mRNA upregulation at 1 hour.*

Response: We apologize for the misunderstanding, but all data on IL-2 mRNA gene expression were performed at 6 h from stimulation, as indicated in the legend to figures. To better clarify this point, and according to the reviewer's suggestions, we performed the analysis of cytokine gene expression at longer time points, in human CD4⁺ T cells stimulated with agonistic or superagonistic anti-CD28 Abs. As reported in the new Supplementary Fig. 1, almost cytokines reached a peak of expression after 6 h and return to a basal level after 24 h from stimulation (Supplementary Fig. 1a-c). By contrast, TNFα expression peaked at 1 hour and returned to a basal level after 6 hours from stimulation (Supplementary Figure 1d).

Reviewer #2

Point 1.

Reviewer: *It would be important to show that human CD28 expressed in a mouse T cell induces pro-inflammatory cytokines and vice versa. This is important to establish the basic phenomenon as the alternative hypothesis is that this can be explained by differences in the antibody.*

Response: We thank the reviewer for the helpful suggestions. We have already shown that mCD28 was not able to induce any pro-inflammatory cytokine gene expression when expressed in human Jurkat cells (see Supplementary Fig. 4d-f). According to the reviewer's suggestion, we also expressed human CD28 in EL-4 mouse T cell lines and we analysed cytokine gene expression following stimulation with agonistic (CD28.2) or superagonistic (ANC28.1) anti-CD28 Abs. As shown in Supplementary Fig. 4, stimulation of EL-4 cells with anti-human CD28 Abs efficiently up-regulated IL-6 (g) gene expression.

Point 2.

Reviewer: *Reconstitution of the CD28 negative Jurkat does not show the level of CD28 expression in the untransfected cell line, only the antibody control is shown.*

Response: According to the reviewer's suggestion, we added the FACS analysis of CD28-negative CH7C17 Jurkat T cell line stained with phycoerythrin (PE) conjugated isotype matched (Iso PE) or anti-CD28 (CD28 PE) Abs (see new Fig. 3b).

Point 3.

Reviewer: *If this is a specific effect on pro inflammatory cytokine, why is there also a defect in NFAT and IL2 production?*

Response: The pivotal role of Nck and actin polymerization in TCR/CD28-mediated signal transduction and activation has been extensively described by several groups (see for review reference N. 20, 22, 25, 42, 43). Thus, the impairment of Nck recruitment by P212A mutant likely affects NF-AT transcriptional activity. The new results on the inhibitory effects of Nck R311Q mutant or cythocalasin D on TCR/CD28-induced NF-AT activation and IL-2 gene expression (see new Fig. 10) confirm these data.

Point 4.

Reviewer: *Again, the data showing distinct complexes formed with human vs mouse CD28 needs more work. Vav binds to both mouse and human but GRB2 binds only to mouse and not human. Lastly, Nck binds to human but not mouse. Does the addition of proline to mCD28 recapitulate the phenotype, meaning loss of GRB2 binding and addition of NCK binding in mouse cells?*

Response: We agree with the reviewer that the characterization of CD28, Vav, Nck and Grb2 complexes require more biochemical experiments and a detailed proteomic work that needs longer time. Indeed, YAPP sequence is an abundant phosphotyrosine site that, as recently demonstrated by Tian et al., has the ability to bind several distinct signaling proteins following CD28/B7 interaction (Tian et al. Proc Natl Acad Sci USA 2015). Thus, the clarification of the nature of CD28/Nck/Vav1 complexes requires the generation of different mutated constructs and a deeper analysis of binding stoichiometry that we are doing. We believe that the *ex-vivo* data on the loss of CD28/Nck binding in primary T cells together with confocal results and the new data on the dominant effects of Nck mutated in its SH2 domain strongly support a role of the YAPP sequence in recruiting Nck and in mediating the differential costimulatory activities of CD28 in human and mouse. For these reasons, we prefer to concentrate our work on Vav1 and Nck.

Point 5.

Reviewer: *The images in figures 6 is not completely convincing. The image of the chimera CD28A210P looks like minimal recruitment. Why doesn't it reconstitute similar to human if this is the only difference in the sequence? The NCK recruitment images look much more convincing.*

Response: The statistical analyses performed on CD28WT, chimera CD28WT and chimera CD28A210P conjugates indicate a strong significant decrease ($p < 0.01$) of Nck (75%), Vav (70%) and F-actin (65%) in Chimera CD28WT compared to hCD28WT. By contrast, no significant differences were observed in Chimera CD28A210P. We added the RRI mean values for each group in the legend to Figures 6 and 7. Moreover, we also performed a further analysis by co-expressing Vav and Nck in hCD28, Chimera CD28WT and Chimera CD28A210P. The results in the new Supplementary Fig. 7, confirmed the lack of co-recruitment of Nck and Vav in Chimera CD28WT, but not in Chimera CD28A210P.

Point 6.

6a. Reviewer: *The idea that CD28 binds to the NCK SH2 domain would require that the tyrosine in the proline motif be phosphorylated. Is there any evidence for this?*

Response: Recent combinatorial proteomic data from Tian et al. evidenced that Y209 within the proline motif is an abundant phosphotyrosine site that has the ability to bind several signaling proteins following CD28/B7 interaction (Tian et al. Proc Natl Acad Sci USA 2015). Consistently, we found that Y²⁰⁹F mutation induces a strong reduction of CD28 tyrosine phosphorylation (new Supplementary Fig. 9b, middle panel), without affecting the tyrosine phosphorylation of Y191, as demonstrated by the unaffected co-precipitation of p85 subunit of PI3K (new Supplementary Fig. 9b, upper panel).

6b. Reviewer: *Also, the data suggests that Vav and NCK are competing for the same binding site.*

Response: We agree with the reviewer that Nck and Vav may compete for the same binding site, because both proteins have an SH2 domain that may bind the phosphorylated Y²⁰⁹APP²¹² within the C-terminal proline rich motif of CD28. To clarify this issue, we performed further experiments by expressing Vav1 and Nck alone or in combination and we analysed their association with CD28. Although, either Vav1 or Nck were able to associate with human CD28 when expressed alone (Supplementary Fig. 8a, b), CD28 preferentially bonds to Nck when both Nck and Vav1 were co-expressed (Supplementary Fig. 8c). We believe that these results, together with the confocal data showing that an intact SH2 domain of Nck, but not Vav1, was required for actin polymerization, Nck and Vav1 recruitment (Fig. 8, 9), and the new data on the inhibitory effects of Nck SH2 mutant on CD28-mediated pro-inflammatory and costimulatory signaling properties (Fig. 10), support a role for the SH2 domain of Nck in linking human CD28 to the cytoskeleton reorganization events necessary for the activation of downstream signaling pathways and biological functions.

6c. Reviewer: *Could differences in the amount of both proteins in mouse and human cells be a potential explanation for differential binding?*

Response: We thank the reviewer for his/her helpful suggestion. We compared the levels of Vav1 and Nck between human and mouse CD4⁺ T cells. No differences were observed between the two species (see new Fig. 5e).

Point 7.

7a. Reviewer: *The assumption is that this mediated by the VAV SH3 domain but this isn't shown.*

Answer: The association of Nck with Vav has been initially demonstrated by Barda-Saad et al., who also identified the C-terminal SH3 domain of Nck and the N-terminal SH3 domain of Vav1 as

responsible for their reciprocal association (Barda-Saad et al *EMBO J* 29:2315-2328.). We have previously confirmed a cooperative interaction of Vav1 with Nck (see Fig. 7 of Muscolini et al *J. Immunol.* 2015). According to the reviewer's suggestions, we performed biochemical-co-precipitating experiments by using either a Vav1 mutant lacking all the SH3 and SH2 domains or a Nck mutant in its SH3.3 domain. In contrast to data from Barda-Saad, we did not observe any variation in the amount of Vav1 co-precipitating with Nck, as shown in the Fig.1 for referee use only (RFig1). Since, we cannot exclude technical limits of the co-precipitation experiments especially because both Vav1 and Nck contain several binding domains that may interact each other when the proteins are overexpressed, we prefer just discuss these alternative possibilities without showing contrasting data.

RFig 1: (a-c) CH7C17 Jurkat cells were transfected with Vav1-myc, or Vav deleted in all SH3 and SH2 domains (Δ SH), or HA-Nck WT, or HA-Nck mutant in the SH3.3 domain (W235K) alone or in combination. Anti-HA immunoprecipitations (IP) were performed on total lysates (TL). Anti-HA, anti-Vav or anti-myc blottings were performed on anti-CD28 IP or TL. Data are representative of three independent experiments

7b. Reviewer: *Another important control would be to show that the SH3 domains of NCK are "not" mediating the recruitment.*

Answer: We apologize for the misunderstanding. In Figure 8, both Nck WT and Nck mutant in SH2 domain (R311Q) contain intact SH3 domains. Thus, the inhibitory effects exerted by the Nck R311Q mutant in its SH2 exclude an involvement of Nck SH3 domains. According to the reviewer's suggestion, we also performed confocal analysis by using a Nck mutant in its SH3 domain (W149K). As shown in the new Figure 8, mutation of the SH3 domain of Nck did not affect neither Nck, nor Vav1 recruitments nor actin polymerization.

Additional corrections: For a mistake, in the previous version of Figure 2, panel e was marked IFN γ instead IL-1 β . We corrected the mistake and in the revised Figure 2, both data have been added (see new Fig. 2e, f)

Thank you for these constructive comments that have allowed us to improve the manuscript. We hope that these changes will fit to *Nature Communications*, allowing the manuscript publication.

Reviewers' comments:

Reviewer #1 (Remarks to the Author):

In this revised manuscript, the authors have added new data to support their hypothesis that proline 212 (P212) of human CD28 protein is crucial for super-agonist anti-CD28-induced inflammatory cytokine production in human T cells. The new figures 4i-4m are critical data demonstrating the sufficiency of P212 for human CD28 induced NF- κ B activation and cytokine secretion. In the new figure 10, by overexpressing mutant Nck or treating cells with cytochalasin D, the authors provide evidence for the requirement of the SH2 domain of Nck and actin cytoskeleton for CD28 induced cytokine production.

However, there are some data presented and cited that argue against a direct interaction of Nck with the Pro212 in CD28. The authors should reconsider whether this interaction is indeed direct:

1. In a paper by Tian et al., PNAS, that the authors cite for evidence for phosphorylation of Y207, an interactome with isolated phosphorylated cytoplasmic peptides of CD28 fails to detect an interaction with Nck among the 28 proteins that do interact with CD28.

2. Nck is said to be equivalently expressed in mouse and human T cells. Yet, the authors use a reduced exposure of 5c (mouse T cell TL and ippts) in comparison with 5a (human T cells). It seems that there might be faint Nck band in 5c that would be more apparent with comparable exposures between human and mouse total lysates and ippts.

3. Nck interacts with Vav, SLP-76, and ADAP among many more.

A direct interaction was not established here.

Reviewer #2 (Remarks to the Author):

The revised manuscript largely satisfies my previous concerns. In particular, the expression of mouse CD28 in human cells, and human CD28 in mouse cells and associated mutants is much more convincing that the P to A change between mouse and human is relevant. The explanation that this is due to Sh2 binding of VAV to the motif is also a satisfying explanation. My only minor issue is that the authors reference Tian et al as evidence that Y209 is phosphorylated. It would have strengthened the manuscript to show that the tyrosine is also important for CD28 signaling.

Point by point reply to reviewers' comments

We are grateful to the reviewers for their further suggestions and advices that have been carefully addressed to improve several aspects of the manuscript. We hope that our corrections satisfy their requests.

Reviewer #1:

Point 1

Reviewer: In a paper by Tian et al., PNAS, that the authors cite for evidence for phosphorylation of Y207, an interactome with isolated phosphorylated cytoplasmic peptides of CD28 fails to detect an interaction with Nck among the 28 proteins that do interact with CD28.

Response: We agree with the reviewer that our data do not provide sufficient evidences for a direct interaction of the SH2 Nck with CD28 and we discussed alternative possibilities in the revised manuscript. Concerning the data by Tian et al., the interactome results were obtained following stimulation of Jurkat cells with SEE-pulsed APC, where both TCR and CD28 were co-engaged. In this experimental condition, CD28 may recruit different signaling mediators compared to CD28 stimulation alone. For instance, when TCR is engaged, Nck can be recruited to TCR by either a direct Nck association with CD3 ϵ or via LAT/SLP-76. It would be interesting to compare the interaction profiles of CD28 binding proteins in the presence or absence of TCR engagement. These alternative possibilities have been further discussed in the revised manuscript (see page 16, yellow highlighted text)

Point 2.

Reviewer: Nck is said to be equivalently expressed in mouse and human T cells. Yet, the authors use a reduced exposure of 5c (mouse T cell TL and ippts) in comparison with 5a (human T cells). It seems that there might be faint Nck band in 5c that would be more apparent with comparable exposures between human and mouse total lysates and ippts.

Response: The amounts of human and mouse total lysates charged in panels 5a and 5c were not equal, because they were just used to identify the position of Nck. Thus, also to longer and saturating exposures, they did not reach equal signals. However, according to the reviewer's suggestion, we modify the figure 5c by adding a longer saturating exposure of both mouse TL and IP (see the new Fig 5c). No significant levels of Nck were identified in mouse CD28 IP compared to human T cells.

Point 3.

Reviewer: Nck interacts with Vav, SLP-76, and ADAP among many more. A direct interaction was not established here.

Response: We agree with the reviewer that more extensive experiments are required to clarify and better characterize the molecular basis and the dynamic of CD28/Nck association. Interestingly, in the paper of Tian et al., SLP-76 has been identified among the preferential binding proteins of CD28 PpYAPP motif. Moreover, the relevance of SLP-76 in both recruitment of the Nck/Vav1-WASp complex, via the Nck SH2 domain, and actin reorganization together with its previously shown cooperation with Vav1 in TCR-independent CD28 signaling, likely suggests a role of SLP-76 in linking CD28 to both Nck and Vav1 signaling functions. We thank the reviewer for these helpful suggestions that have been discussed in the revised manuscript (page 16 and 17, yellow highlighted text).

Reviewer #2 (Remarks to the Author):

The revised manuscript largely satisfies my previous concerns. In particular, the expression of mouse CD28 in human cells, and human CD28 in mouse cells and associated mutants is much more convincing that the P to A change between mouse and human is relevant. The explanation that this is due to Sh2 binding of VAV to the motif is also a satisfying explanation. My only minor issue is that the authors reference Tian et al as evidence that Y209 is phosphorylated. It would have strengthened the manuscript to show that the tyrosine is also important for CD28 signaling.

Response: We thank the reviewer for suggestions and we are generating a single Y209F mutant for better clarifying the dynamic of CD28/Nck/Vav association. However, we have previously demonstrated that the tyrosine residues (Y²⁰⁶QPY²⁰⁹APP) within the C-terminal proline rich motif of human CD28 are crucial for cytoskeleton reorganization and the activation of IKK α and β NF- κ B pathway (Muscolini et al. 2011 Immunol Lett). These data have been discussed in the revised manuscript (see page 15, yellow highlighted text).

REVIEWERS' COMMENTS:

Reviewer #2 (Remarks to the Author):

I'm satisfied with the revised manuscript.